# Dimensional Collapse in VQVAEs: Evidence and Remedies

**Jiayou Zhang**[1], **Yifan Shen**[1], **Guangyi Chen**[1], **Le Song**[1,2], **Eric P. Xing**[1,2,3]

[1]MBZUAI    [2]GenBio AI    [3]Carnegie Mellon University

{jiayou.zhang, yifan.shen, guangyi.chen, le.song, eric.xing}@mbzuai.ac.ae

## Abstract

Vector-Quantized Variational Autoencoders (VQVAEs) have enabled strong performance in generative modeling by mapping continuous data to learnable codes. In this work, we identify a surprising yet consistent phenomenon that we term *dimensional collapse*: despite using high-dimensional embeddings, VQVAEs tend to compress their representations into a much smaller subspace, typically only 4 to 10 dimensions. We provide an in-depth analysis of this phenomenon and reveal its relation to model performance and learning dynamics. Interestingly, VQVAEs naturally gravitate toward this low-dimensional regime, and enforcing higher-dimensional usage (e.g., via rank regularization) could lead to degraded performance. To overcome this low-dimensionality limitation, we propose **Divide-and-Conquer VQ (DCVQ)**, which partitions the latent space into multiple low-dimensional subspaces, each quantized independently. By design, each subspace respects the model's preference for low dimensionality, while their combination expands the overall capacity. Our results show that DCVQ overcomes the inherent dimensional bottleneck and achieves improved reconstruction quality across image datasets.

## 1 Introduction

Learning discrete representations has become a key building block in modern generative modeling, with Vector-Quantized Variational Autoencoders (VQVAEs) [18] standing out as a popular approach. VQVAEs have demonstrated strong empirical success across domains such as images, videos, audio, and protein structures [4, 17, 20, 2, 19, 6], enabling efficient and flexible generative pipelines. At the core of VQVAEs is the idea of discretizing a continuous latent space into learnable codes with code associated with an embedding.

Prior work has primarily focused on *codebook collapse*, where only a subset of codebook entries is utilized. In contrast, we uncover a different and pervasive phenomenon, which we term **dimensional collapse**: despite using high-dimensional embeddings, VQVAEs tend to compress their representations into a much smaller subspace, typically around 4 to 10 dimensions (Figure 1a). This behavior consistently appears across a wide range of pretrained VQVAE models spanning domains and architectures.

To better understand this behavior, we conduct large-scale controlled experiments varying dataset, architecture, and hyperparameters. We observe a surprising U-shaped relationship between effective dimensionality and validation loss: performance improves as the dimension increases, but worsens beyond a certain optimal point (Figure 1b). Notably, the optimum lies in a low-dimensional regime, highlighting the model's inherent preference and a limitation in its expressiveness. A natural goal is to shift this optimum toward higher dimensionality and lower loss (Figure 1c).

Achieving this goal requires understanding why the model favors low-dimensional solutions. We find that collapse originates in the quantizer early in training, with the commitment loss pulling the encoder to match this structure and reinforcing a self-reinforcing loop. Beyond training dynamics, we

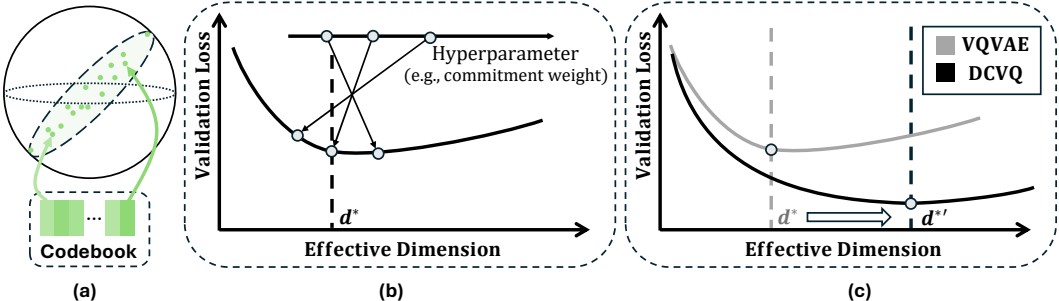

Figure 1: Illustration of dimensional collapse in VQVAEs. **(a)** Codebook entries lie in a low-dimensional subspace despite being in a high-dimensional embedding space. **(b)** Validation loss follows a U-shaped curve as a function of effective dimension, with the optimum at a low dimension $d^*$. Hyperparameters such as the commitment loss weight control the effective dimension. **(c)** Our method, DCVQ, shifts the optimum to a higher effective dimension $d^{*'}$ and achieves lower loss.

show that this behavior stems from a structural bias of quantization itself, which favors high-variance directions and contracts the representation's effective dimension. While rank regularization can counteract this bias by increasing dimensionality, it consistently degrades performance, indicating that naively enforcing rank is ineffective.

Motivated by these insights, we propose **Divide-and-Conquer VQ (DCVQ)**, a simple yet effective architectural modification that addresses collapse at its source: the quantizer. DCVQ partitions the latent space into multiple low-dimensional subspaces, each quantized independently. This design respects the model's natural preference for low-rank structure while scaling capacity through parallel subspaces. As a result, DCVQ breaks the intrinsic low-dimensional limit and achieves improved reconstruction quality and effective dimensionality across datasets.

**Contributions.** Our main contributions are:

- We identify and systematically characterize **dimensional collapse** in VQVAEs, where latent representations occupy a surprisingly low-dimensional subspace.

- We show that collapse arises not only from training dynamics but from a structural bias of quantization itself, which favors high-variance directions and contracts the representational spectrum.

- Through controlled experiments, we reveal a U-shaped relationship between effective dimension and performance, and show that rank regularization fails to resolve collapse, often degrading performance.

- We propose **DCVQ**, which partitions the latent space into multiple independently quantized low-dimensional subspaces, effectively breaking the U-shape limit.

- We validate DCVQ across datasets and architectures, showing consistent gains in reconstruction quality and dimensional utilization.

## 2 Background and problem setup

In this section, we formalize the VQVAE framework, introduce key concepts related to dimensional collapse and describe how we estimate the effective dimensionality.

### 2.1 Vector-Quantized Variational Autoencoders (VQVAEs)

A VQVAE consists of three main components: an encoder $\mathcal{E}$, a codebook $\mathcal{C} = \{e_i\}_{i=1}^{K}$, and a decoder $\mathcal{D}$. Given an input $x$, the encoder produces a continuous latent vector $z = \mathcal{E}(x) \in \mathbb{R}^d$, where $d$ is the **background dimensionality**. This vector is then discretized by replacing it with its nearest codebook entry:

$$\hat{z} = e_{k^*}, \quad \text{where} \quad k^* = \arg\min_k \|z - e_k\|_2$$

If the encoder outputs multiple latent vectors (e.g., patch embeddings), quantization is applied independently to each. The decoder reconstructs the input from the quantized latent: $\hat{x} = \mathcal{D}(\hat{z})$.

The training objective combines a reconstruction loss (e.g., mean squared error) and a *commitment loss* that encourages the encoder output $z$ to stay close to the selected codebook vector $\hat{z}$. When using exponential moving average (EMA) updates [18], the codebook embeddings are updated separately from gradient descent and thus do not appear in the training loss. The resulting training loss is:

$$\mathcal{L} = \|x - \hat{x}\|_2^2 + \beta \|z - \text{sg}[\hat{z}]\|_2^2,$$

where $\text{sg}[\cdot]$ denotes the stop-gradient operator, and $\beta$ controls the strength of the commitment.

## 2.2 Dimensional collapse in VQVAEs

While the background dimensionality $d$ is fixed by model design, the *effective dimensionality* refers to the number of directions in latent space that are meaningfully utilized by the model. In practice, we observe that VQVAEs often use only a low-dimensional subspace of the full latent space, a phenomenon we refer to as **dimensional collapse**.

Formally, given the code vectors $\mathcal{C} = \{e_i\}_{i=1}^K$, we estimate the effective dimensionality using principal component analysis (PCA). The explained variance ratio $\lambda_j$ of each principal component $j$ indicates how much variance is captured along that direction. We define the effective dimension (Eff. Dim) as the minimum number of principal components required to explain a fixed proportion (e.g., 99%) of the total variance:

$$\text{Effective Dim} = \min \left\{ d' : \sum_{j=1}^{d'} \lambda_j > 0.99 \right\}.$$

Unless otherwise noted, we use a 99% variance threshold throughout our experiments.

## 3 Analyzing dimensional collapse

In this section, we perform a comprehensive analysis of *dimensional collapse* in Vector-Quantized VAEs (VQVAEs). We begin by surveying a broad set of pretrained models and observe that collapse emerges as a general and robust phenomenon, across modalities and architectures.

We then conduct large-scale controlled experiments to gain deeper insight into the relationship between effective dimensionality and model performance. These studies reveal a consistent U-shaped curve: while extremely low-dimensional codes hurt expressiveness, forcing higher dimensionality beyond a low-dimensional sweet spot also leads to performance degradation.

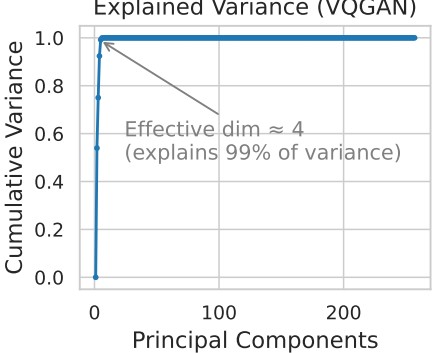

Figure 2: Dimensional collapse in VQGAN (trained on ImageNet). Over 99% of variance is captured by only 4 principal components, despite a 256-dimensional background space.

To explain this behavior, we analyze training dynamics and show that dimensional collapse originates early in the quantization layer, then propagates to the encoder through the commitment loss. This insight is supported by temporal measurements and correlation analysis of key hyperparameters.

Finally, we evaluate whether rank regularization, an intuitive fix, can resolve collapse. Our experiments show that while such regularization increases effective dimensions, it also harms performance, highlighting that collapse reflects the quantizer's training dynamics. These findings motivate the need for an architectural solution, which we present in Section 4.

## 3.1 Collapse in pretrained models

To assess the generality of dimensional collapse, we begin by analyzing a diverse collection of pretrained VQ models spanning multiple domains, including images, video, and protein structures. For each model, we extract the latent codebook embeddings and compute their *effective dimensionality* via Principal Component Analysis (PCA), defined as the number of principal components required to explain 99% of the total variance.

We start with VQGAN, one of the most widely used VQVAE variants. As shown in Figure 2, despite a 256-dimensional background space, over 99% of the variance is captured by just four principal components, which is an extreme case of collapse. The sharp slope in the cumulative variance curve highlights how aggressively the model compresses information into a narrow subspace.

Figure 3 extends this observation across a wider range of models and modalities. Both vanilla VQ-VAE and RQVAE [9] consistently exhibit low effective dimensionality—often below 20—even when the background dimension is 256. This pattern holds across domains, indicating that the collapse is not dataset-specific but instead reflects architectural or training-level constraints.

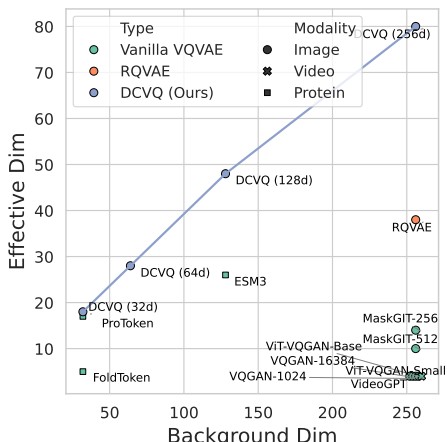

Figure 3: Effective vs. background dimensionality across pretrained VQ models. Vanilla VQVAE and RQVAE consistently underutilize their latent space. Multiple models lie at (256, 4); to improve visibility, their positions are slightly jittered.

## 3.2 VQVAEs's low-dimensional preference: U-shaped performance curve

To systematically study the conditions under which dimensional collapse arises and affects performance, we conduct a large-scale controlled experiment. We train 512 VQVAEs from scratch under diverse combinations of dataset, architecture, and hyperparameter settings, allowing us to isolate key factors influencing effective dimensionality.

### 3.2.1 Experimental Setup

**Datasets.** We use CIFAR-10 [8] and CelebA [12], following standard preprocessing pipelines. CelebA images are resized and center-cropped to $64 \times 64$ resolution. All images are normalized using dataset-specific mean and standard deviation. Our implementation builds on [5] and [21].

**Architectures.** We evaluate both CNN-based and ViT-based encoder-decoder architectures. These are based on VQGAN [4] and ViT-VQGAN [23], respectively. We vary the encoder scale factor ($f$), background dimension, and codebook size. All models apply L2 normalization before quantization.

**Hyperparameters.** Table 1 summarizes the search space. To vary the background dimension without changing the encoder, we insert a projection layer after the encoder and a corresponding inverse projection before the decoder. This setup enables us to isolate the codebook background dimensionality from the encoder and decoder dimensionality.

| Hyperparameter | Values Sampled |
|---|---|
| Codebook size | 512, 1024, 2048, 8192 |
| Background dimension | 3, 4, 6, 8, 16, 32, 64, 128, 256 |
| Commitment loss weight | 0.1, 0.2, 0.5, 0.8, 1.0, 1.2, 1.5, 2.0, 2.5, $\cdots$, 5.0, 5.5, 6.0 |
| EMA decay | 0.5, 0.8, 0.9, 0.95, 0.98 |
| Rotation trick | Enabled / Disabled |
| Dead-code restart threshold | 0.008, 0.032, 0.125, 0.5 |

Table 1: Summary of hyperparameters explored in the large-scale controlled study.

### 3.2.2 Discovering the U-shaped performance curve

We next examine how effective dimensionality influences reconstruction performance across models. Using the 512 configurations described above, we train all models for 100 epochs under consistent training settings to ensure comparability across architectures and hyperparameters.

Figure 4 plots the validation loss as a function of effective dimensionality for different datasets (CIFAR-10, CelebA), model types (CNN, ViT), and encoder scale factors ($f = 4, 16$). Each point represents a trained model, with color indicating codebook size.

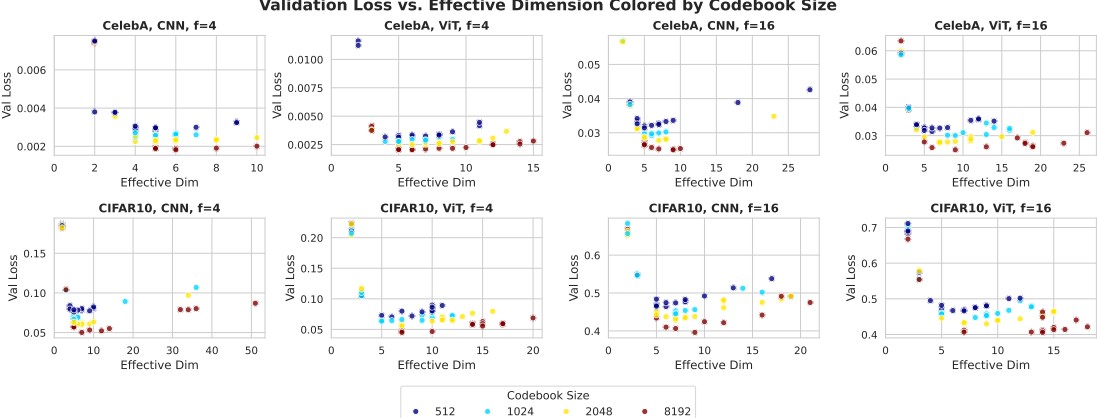

Figure 4: Validation loss (reconstruction MSE) versus effective dimension across datasets, architectures, and scale factors. Color represents codebook size. Within each color group, one can observe a U-shaped curve, indicating a consistent relationship between effective dimension and performance.

Across most settings, we observe a clear U-shaped pattern: models with very low effective dimensionality perform poorly, while performance improves as dimensionality increases up to a certain point. Beyond this point, performance begins to degrade, resulting in a distinct U-shaped curve. For example, on CIFAR-10 (ViT, $f = 4$), models with effective dimensions in the 4–10 range achieve the best performance; higher dimensions offer no gain or even degrade results.

This U-shape suggests the existence of an intrinsic "sweet spot" in latent space usage, typically low but not minimal, where the model achieves optimal reconstruction fidelity. For each model, this corresponds to a fixed effective dimension $d^*$ that it consistently converges to when properly trained. A natural question then arises: how can we shift this optimal effective dimension higher to enhance the model's expressiveness? To answer this question, we first investigate the cause of the collapse.

## 3.3 Understanding the cause of collapse

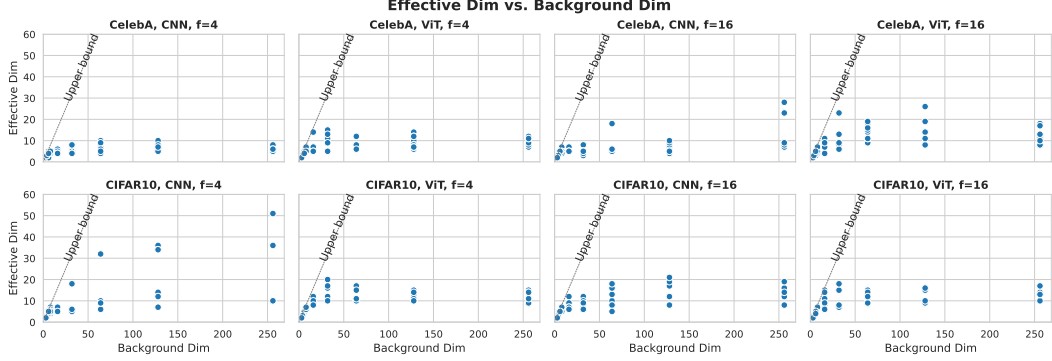

Figure 5: Effective dimension versus background dimension across datasets, architectures, and scale factors. The dashed line indicates the upper bound where effective dimension equals background dimension. As background dimension increases, effective dimension tends to plateau, indicating that higher latent capacity is not utilized.

We begin by examining whether dimensional collapse is simply a consequence of limited latent size. As shown in Figure 5, when the background dimension is small (e.g., 3–6), the effective dimension scales linearly. However, once the background dimension becomes sufficiently large, the effective dimension saturates and diverges from the upper bound. This decoupling suggests that collapse is not caused by latent capacity itself, but rather emerges from internal training dynamics. Notably, this saturation effect occurs across datasets and architectures, implying a general phenomenon.

To gain a deeper understanding of where and when collapse originates, we analyze the training-time dynamics of effective dimensionality. As shown in Figure 6, collapse occurs rapidly—within the first 5–10k training steps—and then plateaus. More importantly, we observe that the codebook consistently

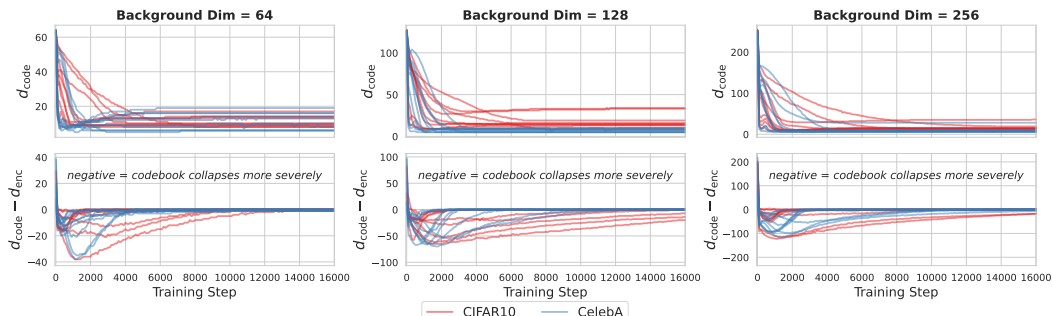

Figure 6: Evolution of effective dimensionality over training steps for different background dimensions ($d = 64, 128, 256$). **Top row:** effective dimensionality of the codebook embeddings ($d_\text{code}$). **Bottom row:** difference between codebook and encoder dimensionality ($d_\text{code} - d_\text{enc}$), where negative values indicate that the codebook collapses more severely than the encoder. Collapse occurs rapidly in early training and then stabilizes, with the encoder gradually adapting to the collapsed codebook.

collapses earlier than the encoder. The bottom row shows the dimensionality gap ($d_\text{code} - d_\text{enc}$)[1]: it is initially negative, revealing that the quantizer underutilizes dimensions first. Over time, the encoder's dimensionality drops as it conforms to the collapsed codebook.

These results indicate that collapse is not a failure of encoder expressiveness, but is instead induced by early-stage quantizer behavior. This in turn suggests that collapse originates from the training dynamics of the quantization mechanism itself.

To further validate this hypothesis, we analyze correlations between hyperparameters and final effective dimensionality (Table 2). Among all factors, the **commitment loss weight** shows the strongest and most consistent negative correlation across background dimensions. This aligns with the temporal findings: the commitment loss acts as the conduit through which early quantizer collapse is propagated to the encoder.

Taken together, these results support a coherent causal explanation: dimensional collapse originates in the quantizer due to unstable training dynamics, and is reinforced by the commitment term that pulls encoder outputs toward a degenerate codebook.

Table 2: Pearson correlation between effective dimensionality and hyperparameters, grouped by background dimension. Commitment loss weight shows the strongest and most consistent correlation.

| Hyperparameter | Background Dim | | | | | | | Avg. |
|---|---|---|---|---|---|---|---|---|
| | **6** | **8** | **16** | **32** | **64** | **128** | **256** | |
| Commitment Loss Weight | -0.45 | **-0.72** | **-0.79** | **-0.68** | **-0.50** | **-0.62** | -0.40 | **-0.60** |
| Codebook Size | **0.57** | 0.19 | 0.17 | 0.49 | 0.39 | 0.55 | **0.59** | 0.42 |
| Rotation Trick | 0.29 | 0.19 | 0.13 | -0.02 | 0.41 | 0.39 | 0.44 | 0.26 |
| Code Restart Threshold | -0.10 | -0.09 | 0.06 | -0.09 | -0.08 | -0.01 | 0.15 | -0.02 |
| EMA Decay | -0.04 | 0.03 | 0.10 | 0.07 | 0.05 | 0.03 | -0.03 | 0.03 |

### 3.4 Quantization bias as a structural cause of collapse

While the previous analysis shows that collapse originates in the quantizer, the underlying cause remains to be explained. We hypothesize that this behavior is not incidental but arises from an inherent bias of the quantization process itself. Vector quantization in VQ-VAE can be viewed as an online form of $k$-means, with codebook entries as centroids. Classical analyses relate $k$-means to PCA by showing that the continuous relaxation of the $k$-means objective is a trace maximization problem whose global optima are given by the top eigenvectors of the covariance/Gram matrix; equivalently, principal components arise as continuous solutions to the discrete cluster-indicator problem [27, 3]. This linkage implies that K-means effectively operates within the leading-variance

---

[1]$d_\text{enc}$ is computed from the encoder outputs of the current batch.

subspace, which helps explain our empirical finding that low-variance directions are attenuated after quantization. This bias naturally leads to a contraction of the representation's effective dimension.

Figure 7 illustrates this behavior in a controlled synthetic setting. We generate high-dimensional Gaussian data with a prescribed covariance spectrum and apply $k$-means clustering to learn 512 centroids, then compare the covariance eigenvalues of the data and centroids to quantify the change in effective rank (code provided in Sec. G). The accompanying visualization shows that smaller eigenvalues shrink markedly after clustering, while a projection onto the first and last principal components reveals that centroids spread along the high-variance direction but collapse along the low-variance axis.

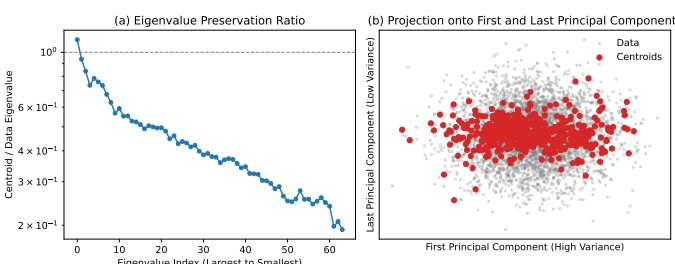

Figure 7: Synthetic illustration of quantization bias via $k$-means clustering. (**a**) The ratio between centroid and data eigenvalues (log scale) shows that clustering preserves high-variance directions while attenuating low-variance ones. (**b**) A projection onto the first and last principal components confirms this effect visually: centroids (red) align with the high-variance axis, whereas variation along the low-variance direction is largely lost.

This geometric contraction mirrors the theoretical link between $k$-means and PCA—where the relaxed $k$-means solution lies in the subspace spanned by leading eigenvectors [27, 3]—and together they indicate that quantization inherently biases representations toward low-dimensional, high-variance subspaces, offering a structural explanation for dimensional collapse.

In summary, both theoretical and empirical evidence suggest that dimensional collapse is not a pathological optimization artifact but a structural outcome of the quantization process itself. By favoring directions of high variance, quantization inherently compresses the representational spectrum, even when the encoder produces high-rank features.

## 3.5 Failure of Rank Regularization

Given that collapse originates in the quantizer and is reinforced by training dynamics, one might wonder whether it can be mitigated by explicitly encouraging high-rank encoder outputs. To investigate this, we tested a wide range of rank-promoting strategies applied directly to the encoder outputs, including KoLeo regularization [16, 14], Barlow Twins [25], VICReg [1], Spectrum Hinge, and Rank Hinge. These methods address collapse through embedding spreading, redundancy reduction, covariance spectrum shaping, and rank thresholding. Notably, KoLeo, Barlow Twins, and VICReg originate from the self-supervised learning literature, where they are designed to prevent representational collapse. A summary of these methods and their corresponding objectives is provided in Table 7.

Table 3 summarizes the quantitative effects of these regularizers on both CIFAR10 and ImageNet. Across all methods, stronger regularization consistently increases the effective dimensionality of encoder features but also leads to higher reconstruction loss, confirming that the apparent rank expansion occurs primarily before quantization and does not translate into better post-quantization performance.

These results reinforce a key insight: simply increasing the effective dimension does not improve performance. On the contrary, forcing high-rank outputs from the encoder undermines the model's natural dimensional biases and yields lower-quality latent representations. Importantly, encoder-side regularization fails to address the root cause of collapse: the behavior of the quantization layer.

This limitation highlights the need to intervene directly at the quantizer level, motivating a strategy that respects the model's low-dimensional preference while expanding the latent capacity. We introduce such an approach, **DCVQ**, in the next section.

Table 3: Validation reconstruction loss and effective dimension under different rank-promoting regularizers. Results are reported for CIFAR10 (CNN, $f$=4) and ImageNet-1k (ViT, $f$=16). For each regularizer, the lowest validation loss and the highest effective dimension are marked with "*".

| Regularizer | CIFAR10 | | | ImageNet-1k | | |
|---|---|---|---|---|---|---|
| | Weight | Val Loss | Eff. Dim | Weight | Val Loss | Eff. Dim |
| None | | 0.081 | 9 | | | |
| KoLeo | 0.001 | 0.081* | 9 | | | |
| | 0.01 | 0.082 | 10 | | | |
| | 0.1 | 0.106 | 27* | | | |
| Barlow Twins | 0.0001 | 0.081* | 10 | 0.0001 | 0.173* | 14 |
| | 0.001 | 0.093 | 31 | 0.001 | 0.186 | 41 |
| | 0.01 | 0.105 | 78* | 0.01 | 0.205 | 91* |
| | 0.1 | 0.087 | 49 | 0.1 | 0.182 | 58 |
| VICReg | 0.1 | 0.081 | 9 | 0.1 | 0.172 | 9 |
| | 1 | 0.080* | 10 | 1 | 0.164* | 11 |
| | 10 | 0.082 | 11 | 10 | 0.169 | 13 |
| | 100 | 0.085 | 12* | 100 | 0.180 | 20* |
| Spectrum Hinge (hinge=0.01) | 0.1 | 0.080* | 9 | 0.1 | 0.158* | 9 |
| | 1 | 0.081 | 9 | 1 | 0.168 | 11 |
| | 10 | 0.089 | 49 | 10 | 0.181 | 63 |
| | 100 | 0.129 | 111* | 100 | 0.247 | 137* |
| Spectrum Hinge (hinge=0.1) | 0.1 | 0.080* | 9 | | | |
| | 1 | 0.081 | 10 | 1 | 0.171* | 11 |
| | 10 | 0.084 | 11 | 10 | 0.183 | 15 |
| | 100 | 0.085 | 12* | 100 | 0.186 | 17* |
| Rank Hinge (hinge=64) | 0.001 | 0.081 | 9* | 0.001 | 0.173* | 9* |
| | 0.01 | 0.081 | 9* | 0.01 | 0.173* | 9* |
| | 0.1 | 0.080* | 9* | 0.1 | 0.173* | 9* |
| | 1 | 0.081 | 9* | 1 | 0.173* | 9* |
| Rank Hinge (hinge=128) | 0.001 | 0.080* | 9* | 0.001 | 0.166 | 9* |
| | 0.01 | 0.082 | 9* | 0.01 | 0.163* | 9* |
| | 0.1 | 0.080* | 9* | 0.1 | 0.168 | 9* |
| | 1 | 0.081 | 9* | 1 | 0.166 | 9* |

## 4 DCVQ: Divide-and-conquer quantization for dimensional collapse

### 4.1 Motivation

Our analysis of dimensional collapse reveals a surprising yet consistent trend: model performance, as measured by reconstruction quality, first improves and then degrades as the effective latent dimension increases. Critically, the optimal performance typically emerges at a low effective dimension.

This finding suggests that VQVAEs intrinsically prefer operating in low-dimensional latent subspaces. Forcing the model to uniformly utilize a higher-dimensional latent space, such as through explicit rank regularization, would degrade the performance. Therefore, a more natural strategy is to *embrace* the low-dimensional preference rather than fighting against it.

However, solely operating at such low effective dimensions severely limits the total information capacity of the latent space. To balance these competing needs—*respecting low-dimensional structure* while *scaling total capacity*—we propose **DCVQ**.

### 4.2 DCVQ architecture

Divide-and-Conquer Vector Quantization (DCVQ) is a simple architectural modification to VQVAE that addresses dimensional collapse by splitting the latent space into multiple independently quantized

subspaces. This design reflects the intrinsic preference of VQVAEs for low-dimensional structure while scaling total capacity through parallelism.

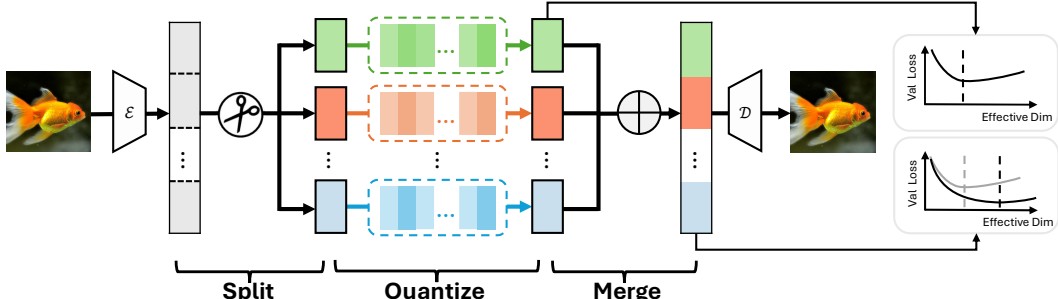

Figure 8: DCVQ divides the encoder output into multiple low-dimensional subspaces and quantizes each independently. The quantized subspaces are then merged via a direct sum (concatenation) and passed to the decoder. This divide-and-conquer strategy enables high total capacity while preserving the model's preference for low-dimensional structure.

**Divide Step.** Given an encoder output $z \in \mathbb{R}^d$, DCVQ partitions it into $N$ equally sized subspaces, each of dimension $d_s$, with $d = N \cdot d_s$:

$$z = [z^1, z^2, \ldots, z^N], \quad z^i \in \mathbb{R}^{d_s}$$

This decomposition enables each subspace to operate independently in a low-dimensional regime close to the model's optimal effective dimension ($d_s \approx d^*$). Based on prior observations, we use $d_s = 8$.

**Conquer Step.** Each subspace $z^i$ is quantized separately using its own codebook $\mathcal{C}_i$:

$$\hat{z}^i = \text{Quantize}(z^i, \mathcal{C}_i)$$

The final latent representation is the concatenation of the quantized subspaces:

$$\hat{z} = [\hat{z}^1, \hat{z}^2, \ldots, \hat{z}^N] \in \bigoplus_{i=1}^{N} \mathbb{R}^{d_s}$$

Concatenation forms a *direct sum* of subspaces. By linear algebra, the rank of the concatenated space equals the sum of the ranks of the individual subspaces. Thus, stacking more quantizers leads to a linear increase in effective dimension. This allows the model to achieve high expressivity without violating its low-rank preference.

# 5    Experiments

## 5.1    Comparison with vanilla VQVAEs

We compare DCVQ with standard VQVAE models across datasets (CelebA, CIFAR10) and architectures (CNN, ViT), with results shown in Figure 9. Here, we fix the codebook size to 512 entries for all models and consider only the setting with $f = 16$. We vary the subspace dimensionality $d_s$ between 8 and 32, with $d_s = 8$ assumed to be near the optimal for individual codebooks.

Across all settings, we observe that DCVQ achieves lower validation loss at significantly higher effective dimensions, demonstrating that our goal in Figure 1c is realized: the performance curve shifts toward both higher dimensionality and lower loss. This confirms that DCVQ overcomes the low-dimensional bottleneck of vanilla VQ-VAEs and scales more effectively. The gray points represent vanilla VQVAEs, which consistently

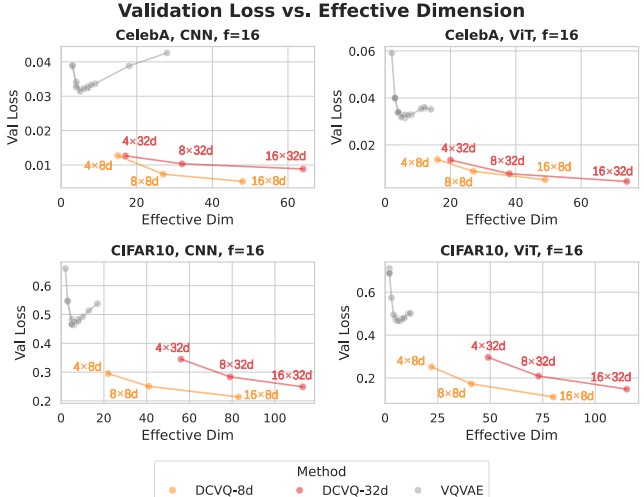

Figure 9: Comparison with vanilla VQVAEs. Text annotations are $N \times d_s$ (i.e., #Quantizers $\times$ Subspace Dim).

operate in a narrow low-dimensional regime. In contrast, DCVQ variants span a broader range of effective dimensions while achieving better loss. Notably, models using $d_s = 32$ exhibit higher validation loss than those using $d_s = 8$, supporting the choice of $d_s = 8$ as a near-optimal subspace dimension. These results demonstrate that DCVQ enables higher-capacity representations without sacrificing reconstruction quality.

## 5.2 Comparison with RQVAEs

Since DCVQ employs multiple codebooks, we compare it against Residual Quantization (RQ, also known as RQVAE) [9], a method that also leverages multiple quantizers.

We evaluate reconstruction performance on ImageNet-256 using reconstruction Fréchet Inception Distance (rFID) as the metric. All models use a convolutional encoder-decoder architecture from the RQ-VAE codebase.

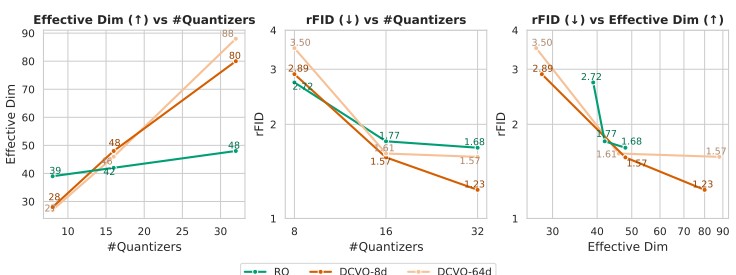

Figure 10: Effective dimension and reconstruction FID (rFID) across different quantizer counts. We vary the number of quantizers while keeping other hyperparameters, such as the dimensionality of each quantizer, fixed to study how reconstruction quality evolves.

For DCVQ, the latent vector is partitioned into subspaces of fixed dimension ($d_s \in \{8, 64\}$), each quantized independently. We vary the number of quantizers $N \in \{8, 16, 32\}$, ensuring that the total codebook dimensionality for RQ matches that of DCVQ-8d (i.e., $d = 8N$). The codebook size is fixed at 16,384 across all settings.

Figure 10 shows that DCVQ achieves significantly higher effective dimensionality than RQ as the number of quantizers increases, indicating more expressive latent codes. DCVQ-8d and DCVQ-64d converge to similar effective dimensions despite differing background sizes, consistent with our earlier observation that beyond a threshold, background dimension no longer constrains capacity.

In terms of rFID, DCVQ consistently outperforms RQ when $N > 16$. While RQ quickly saturates, DCVQ continues to improve with more quantizers, especially with $d_s = 8$, achieving higher effective dimension and better reconstruction quality. These results confirm that DCVQ not only expands latent capacity effectively but also translates that capacity into meaningful performance gains.

## 6 Conclusion and Discussion

We study *dimensional collapse* in VQVAEs, a phenomenon where models utilize only a small subspace (typically 4–10 dimensions) regardless of a high background dimension. This limits expressiveness, as increasing the background dimension yields diminishing gains in effective usage.

Our analysis reveals that collapse originates from early codebook degeneration and cannot be resolved by encoder-side regularization alone. To address this limitation, we propose **Divide-and-Conquer VQ (DCVQ)**, which splits the latent space into low-dimensional subspaces, each independently quantized. This respects the model's low-dimensional preference while expanding total expressivity, achieving both low background dimension and high utility.

Experiments across benchmarks show that DCVQ consistently improves both effective dimensionality and reconstruction quality. These results confirm that DCVQ not only expands latent capacity effectively, but also translates that capacity into meaningful performance gains.

**Limitations.** This work is primarily empirical and does not yet offer a theoretical explanation for codebook collapse. Future work should aim to provide theoretical grounding and extend DCVQ to other modalities.

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

# NeurIPS Paper Checklist

The checklist is designed to encourage best practices for responsible machine learning research, addressing issues of reproducibility, transparency, research ethics, and societal impact. Do not remove the checklist: **The papers not including the checklist will be desk rejected.** The checklist should follow the references and follow the (optional) supplemental material. The checklist does NOT count towards the page limit.

Please read the checklist guidelines carefully for information on how to answer these questions. For each question in the checklist:

- You should answer [Yes] , [No] , or [NA] .
- [NA] means either that the question is Not Applicable for that particular paper or the relevant information is Not Available.
- Please provide a short (1–2 sentence) justification right after your answer (even for NA).

**The checklist answers are an integral part of your paper submission.** They are visible to the reviewers, area chairs, senior area chairs, and ethics reviewers. You will be asked to also include it (after eventual revisions) with the final version of your paper, and its final version will be published with the paper.

The reviewers of your paper will be asked to use the checklist as one of the factors in their evaluation. While "[Yes] " is generally preferable to "[No] ", it is perfectly acceptable to answer "[No] " provided a proper justification is given (e.g., "error bars are not reported because it would be too computationally expensive" or "we were unable to find the license for the dataset we used"). In general, answering "[No] " or "[NA] " is not grounds for rejection. While the questions are phrased in a binary way, we acknowledge that the true answer is often more nuanced, so please just use your best judgment and write a justification to elaborate. All supporting evidence can appear either in the main paper or the supplemental material, provided in appendix. If you answer [Yes] to a question, in the justification please point to the section(s) where related material for the question can be found.

IMPORTANT, please:

- **Delete this instruction block, but keep the section heading "NeurIPS Paper Checklist",**
- **Keep the checklist subsection headings, questions/answers and guidelines below.**
- **Do not modify the questions and only use the provided macros for your answers**.


# Contents

# A   Related work

## A.1   VQVAE and its variants

Vector-Quantized Variational Autoencoders (VQVAEs) [18] have emerged as a cornerstone in discrete representation learning, enabling high-fidelity generative modeling across vision [15], audio [26], video [20], and structural biology [19, 6]. The central idea is to discretize a continuous latent space into learnable embeddings, typically via nearest-neighbor assignment to a codebook.

Successive works have aimed to scale and stabilize this framework. VQVAE-2 [15] introduced hierarchical quantization, while VQVAE [9] proposed recursive residual quantization. Innovations like grouped quantization [22], the rotation trick [5], and joint codebook updates in SimVQ [28] further refine training dynamics and representation efficiency. Lookup-Free Quantization (LFQ) [24] and Finite Scalar Quantization (FSQ) [13] offer scalar-level discretization with extremely low latent dimensions (e.g., 6–8), improving code utilization but at the cost of expressivity.

Despite extensive progress in improving generation quality and mitigating codebook collapse, our work focuses on dimensional collapse, a more subtle and previously overlooked issue.

## A.2   Multi-codebook methods

Several recent works have proposed architectural strategies that superficially resemble our approach by partitioning the latent space into independently quantized subspaces. For example, XQ-GAN and IMAGEFOLDER [11, 10] adopt *product quantization* to split the latent space into low-dimensional branches, each quantized separately. However, their primary goal is to align tokens with spatial or semantic features to facilitate autoregressive modeling and reduce sequence length, rather than addressing the underlying structure of the latent space. In contrast, our work is driven by a systematic analysis of *dimensional collapse* in VQVAEs, where high-dimensional embeddings are compressed into narrow subspaces, limiting expressivity.

Other methods, such as RQVAE [9] and grouped quantization [22], improve expressivity by introducing multi-stage or groupwise quantization to better approximate the encoder outputs. Visual AutoRegressive modeling (VAR) [17] further complements this line by redefining generation as a coarse-to-fine, multi-scale prediction task, achieving state-of-the-art results in efficiency and quality. While these approaches adopt multi-codebook designs to enhance generation, our proposed DCVQ leverages them as a means to address latent space underutilization.

## A.3   Codebook collapse and dimensional collapse

Codebook collapse—the underutilization of codebook entries—has long been recognized as a key limitation in VQVAEs [18]. Numerous approaches have been proposed to mitigate this issue, including dead-code replacement [26], exponential moving average (EMA) updates [15], and architectural modifications that improve gradient flow and codebook dynamics [28, 5]. Interestingly, several works [13, 23] have shown that reducing the embedding dimension can lead to improved code usage, suggesting a link between codebook utilization and the underlying structure of the latent space.

However, most of these efforts focus on usage statistics such as codebook perplexity or entropy, without examining the geometry of the latent space itself. A more fundamental issue—*dimensional collapse*—has only recently begun to attract attention. This refers to the observation that, despite

operating in high-dimensional latent spaces, VQVAEs often encode information in a much lower-dimensional subspace. While the term "dimensional collapse" has previously been used in contrastive self-supervised learning [7] to describe degeneracy in feature representations, its occurrence in VQ-based generative models has not been systematically studied.

## B Case study: visualization of each effective dimension

### B.1 Purpose

This section aims to provide an intuitive understanding of what is encoded in each effective dimension of the VQVAE codebook. By visualizing the role of individual principal components (PCs), we explore how semantic information is distributed across the most significant axes of variation in the learned discrete latent space.

### B.2 Methodology

We perform principal component analysis (PCA) on the codebook embeddings to identify the most informative directions. Specifically, we construct a series of reduced codebooks that contain only the first $k$ principal components ($k = 1$ to $6$). Each level of this PCA-reduced codebook represents an increasingly complete approximation of the full latent space.

For a given input image, we extract its discrete token indices using the encoder. We then replace the standard codebook embeddings with their PCA-reduced counterparts before decoding. This allows us to visualize how reconstructions evolve as more effective dimensions are included.

We conduct the experiment on VQGAN. Notably, our analysis reveals that over 99% of the variance in the codebook is captured by the first 4 principal components. Thus, we focus on up to the first 6 PCs to analyze marginal contributions beyond the main informative axes.

### B.3 Visualization setup

The visualizations are arranged in two rows for each example:

- **Top row:** reconstructions using progressively richer PCA embeddings—from PC1 only up to PC6. The baseline input and full VQGAN reconstruction are also included for reference.

- **Bottom row:** the latent activations corresponding to each individual PC, rendered as grayscale images, to show spatial distribution and intensity.

### B.4 Findings

From the visualizations in Figure 11, we observe the following:

- **PC1** captures the global layout or coarse structure of the image. For example, the silhouette or contour of a person is often visible even with only the first PC.

- **PC2 and PC3** encode finer details such as texture and local contrast, improving visual fidelity and definition.

- **PC4 and PC5** introduce color and tone variations, filling in stylistic and chromatic information.

- **PC6** contributes marginally, with minimal visible structure in either the reconstruction or the latent visualization, indicating that it carries negligible additional semantic content.

This experiment supports the interpretation that VQVAE codebooks suffer from a dimensional bottleneck, where only a few directions in the latent space carry most of the meaningful information. This visualization method complements quantitative measures by providing direct human-interpretable evidence of dimensional collapse.

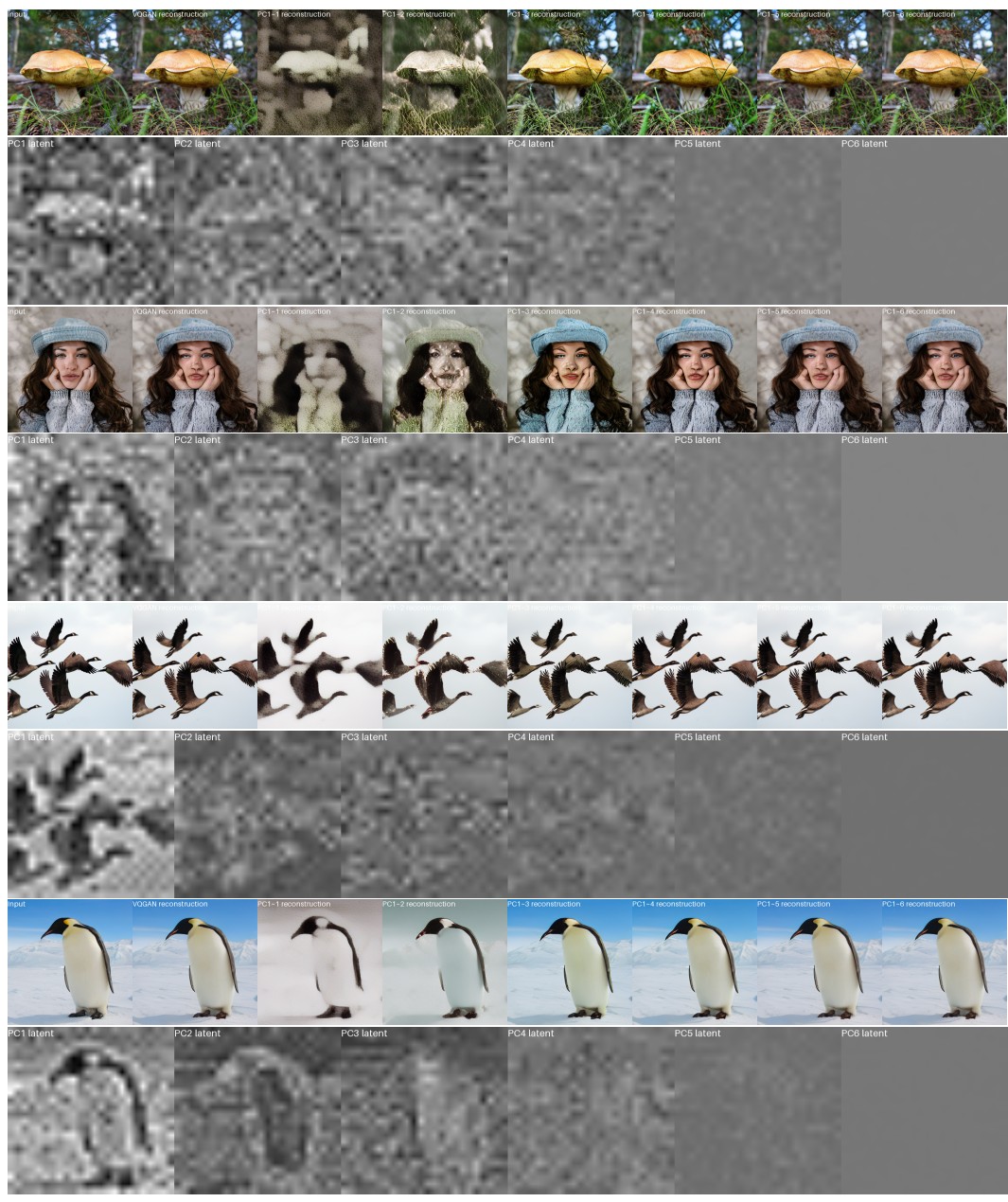

Figure 11: Reconstructions and principal component visualizations for selected examples.

# C   Implementation details

## C.1   Codebase

Our implementation is based on several publicly available GitHub repositories:

- `lucidrains/vector-quantize-pytorch` [21]                    (MIT License)
- `kakaobrain/rq-vae-transformer` [9]                    (Apache 2.0 License)
- `cfifty/rotation_trick` [5]    (No explicit license; used under academic fair use as per official ICLR release)
- `thuanz123/enhancing-transformers`                    (MIT License)
- `CompVis/taming-transformers` [4]                    (MIT License)

Specifically, the training script for VQVAE on CelebA and CIFAR-10 was adapted from `cfifty/rotation_trick`, and the ViT architecture implementation was taken from `thuanz123/enhancing-transformers`. The CNN-based encoder/decoder backbone was based on `CompVis/taming-transformers`. The RQ-VAE training procedure and the training script for ImageNet were derived from `kakaobrain/rq-vae-transformer`. We re-trained RQVAE on our own setup and observed that the performance closely matched the results reported in their paper.

## C.2   Training infrastructure

All experiments were conducted on NVIDIA A100 GPUs with 80GB of memory. Training was performed on a single GPU per run. Although A100s were used, our models did not fully utilize the available memory, making it feasible to train them on lower-end GPUs as well. We employed Weights and Biases (wandb) for experiment tracking and conducted uniform random hyperparameter sweeps. Reported training times (in GPU-hours) are taken directly from the wandb platform.

## C.3   Datasets

We conduct experiments on three datasets: ImageNet-256, CelebA-64, and CIFAR-10. ImageNet images are resized to 256×256 resolution. CelebA images are resized to 64×64, and CIFAR-10 consists of 32×32 images. These datasets cover a range of visual complexity and resolution, allowing us to evaluate the model's behavior across diverse settings.

## C.4   Hyperparameters for CelebA and CIFAR-10

For the experiments on CelebA and CIFAR-10 presented in Section 3 and Section 5 of the main text, we conducted hyperparameter searches over selected variables as described in Table 1 (main text). Other hyperparameters were held fixed across runs. The full configuration used in these experiments is detailed in Table 4. We use a custom learning rate scheduler that linearly warms up, then decays via cosine annealing to a fixed minimum learning rate.

## C.5   Hyperparameters for ImageNet

For the ImageNet experiments in Section 5, we adopt the default hyperparameters provided by the RQVAE codebase, which could be found in Table 5.

## C.6   Architecture Details for CelebA and CIFAR-10

We use a standard VQ-VAE framework composed of an encoder, a vector quantizer, and a decoder. The encoder and decoder architectures differ depending on the model configuration:

- **CNN-based:** We adopt a convolutional encoder-decoder architecture inspired by VQGAN. The model downsamples the input by a factor of 4 or 16, using 3 or 5 convolutional blocks respectively. Each block contains 2 residual layers, and the number of channels is determined by a base width of 128 multiplied by a channel multiplier. The channel multiplier is set to $[1, 2, 4]$ for a downsampling factor of 4, and $[1, 1, 2, 2, 4]$ for a factor of 16. The encoder output is projected to a fixed dimensionality of 256 using a convolutional layer.

| Hyperparameter | Value |
|---|---|
| Batch size | 32 |
| Optimizer | Adam |
| Learning rate | 0.0001 |
| Weight decay | 0.0001 |
| Epochs | 100 |
| Model type | VQVAE with rotation trick |
| Codebook type | cosine |
| Warmup iterations | 3000 |
| Decay iterations | 50000 |
| Stochastic sampling of codes | False |
| Dropout | 0 |
| Seed | 0 |

Table 4: Fixed hyperparameter configuration for the experiments in Section 3.

| Hyperparameter | Value |
|---|---|
| Batch size | 32 |
| Epochs | 10 |
| Optimizer | Adam |
| Learning rate | 4.0e-05 |
| Betas | (0.5, 0.9) |
| Weight decay | 0.0 |
| Learning rate scheduler | fixed learning rate |
| Discriminator loss | Hinge |
| Discriminator start epoch | 0 |
| Discriminator weight | 0.75 |
| Generator loss | Vanilla |
| Perceptual loss weight | 1.0 |

Table 5: Non-architectural hyperparameters used in the experiments in Section 5.

- **ViT-based:** We employ a Vision Transformer (ViT) encoder-decoder architecture for image tokenization and reconstruction. The input image is partitioned into non-overlapping patches of size $f \times f$, where $f$ denotes the patch size (e.g., $f = 4$ or 16). Each patch is embedded via a convolutional projection into a 768-dimensional vector. Fixed 2D sinusoidal positional embeddings are added to the patch sequence, which is then processed by a Transformer encoder comprising 12 layers, each with 12 self-attention heads and an MLP of width 3072. The encoded patch representations are quantized and then passed to a symmetric Transformer decoder to reconstruct the image. This architecture enables global context modeling and adaptive spatial compression.

In both architectures, a linear projection maps the encoder output to the quantizer input space, which allows tunable codebook dimensionality. After quantization, the embeddings are projected back before being passed to the decoder. The quantizer is based on cosine similarity and is updated using exponential moving average (EMA).

## C.7 Architecture Details for ImageNet

For experiments on ImageNet, the settings are similar but we only have CNN-based backbone with $f = 8$. We follow the RQVAE architecture with the following configuration:

- **CNN-based (ImageNet):** We utilize a deep convolutional encoder-decoder architecture designed for high-resolution inputs. The encoder comprises 6 stages, each with 2 residual blocks, totaling 12 residual layers. A base channel size of 128 is used with channel multipliers $[1, 1, 2, 2, 4, 4]$, reaching up to 512 channels in deeper layers. The spatial resolution is reduced by a factor of 8. The encoder output is projected to a 256-dimensional latent

space. A commitment loss with weight 0.25 is applied to encourage consistency between the encoder output and the quantized representation.

## C.8 GPU usage

Table 6 summarizes the GPU usage for the experiment in Section 3. The GPUs we use were A100 80GB.

| Sweep Name | Dataset & Model | GPU Days |
|---|---|---|
| sweep_celeba_cnn_f=4 | CelebA + CNN | 47 |
| sweep_celeba_vit_f=4 | CelebA + ViT | 87 |
| sweep_cifar_cnn_f=4 | CIFAR-10 + CNN | 15 |
| sweep_cifar_vit_f=4 | CIFAR-10 + ViT | 19 |
| sweep_celeba_cnn_f=16 | CelebA + CNN | 34 |
| sweep_celeba_vit_f=16 | CelebA + ViT | 32 |
| sweep_cifar_cnn_f=16 | CIFAR-10 + CNN | 15 |
| sweep_cifar_vit_f=16 | CIFAR-10 + ViT | 17 |

Table 6: GPU time (in GPU-days) consumed by each sweep. Each sweep includes 64 individual runs, totaling 512 runs across all configurations. The sweeps vary by dataset (CelebA or CIFAR-10), model type (CNN or ViT), and downsampling factor ($f = 4$ or $f = 16$).

# D   Details of the rank regularizers

Table 7 summarizes the regularizers in Table 3.

Table 7: Rank-promoting regularization methods applied to encoder outputs. All operate on $\ell_2$-normalized features $x \in \mathbb{R}^{n \times d}$, where $n$ is the batch size and $d$ is the feature dimension. We denote by $\sigma(x._i)$ the feature-wise standard deviation of $x$, by $C(x)$ the covariance matrix of $x$, and by $\{\lambda_i\}_{i=1}^{d}$ the singular values of $x$ obtained from its singular value decomposition (SVD). Batch-normalized $x$ is denoted as $x^{\mathrm{BN}}$.

| Method | Description | Objective |
|---|---|---|
| **KoLeo** [16, 14] | Encourages uniform spreading of features in latent space by maximizing pairwise distances. | $-\dfrac{1}{n} \sum_{i=1}^{n} \log \min_{j \neq i} \|x_i - x_j\|$ |
| **Barlow Twins** [25] | Promotes feature decorrelation by penalizing off-diagonal covariance and enforcing unit variance. | $\sum_{i=1}^{d} \left(C(x^{\mathrm{BN}})_{ii} - 1\right)^2 + \frac{5.1}{1000} \sum_{i \neq j} C(x^{\mathrm{BN}})_{ij}^2$ |
| **VICReg** [1] | Maintains feature variance while reducing cross-covariance between dimensions. | $\frac{1}{d} \sum_{i=1}^{d} \max\left(0, 1 - \sigma(x._i)\right) + \frac{1}{25d} \sum_{i \neq j} C(x)_{ij}^2$ |
| **Spectrum Hinge** | Prevents small singular values by applying a hinge threshold $\tau$. | $\sum_{i=1}^{d} \max(0, \tau - \lambda_i)$ |
| **Rank Hinge** | Enforces a minimum effective rank $r$ by penalizing when the leading $(r-1)$ components explain over 99% of variance. | $\max\left(0, \frac{\sum_{i=1}^{r-1} \lambda_i^2}{\sum_{i=1}^{d} \lambda_i^2} - 0.99\right)$ |

# E   Additional results on Pearson correlations with effective dimensionality

In Table 2 of the main text, we report the average Pearson correlation between effective dimensionality and various hyperparameters, aggregated over different background dimensionalities. Figure 12 provides a more detailed breakdown of these correlations across individual datasets, architecture, scale factors, and background dimensions.

To further disentangle the effect of the Rotation Trick, Tables 8 and 9 report the same correlation analysis separately for cases where the Rotation Trick is enabled and disabled, respectively. The results confirm that the overall trends remain consistent.

Table 8: Correlation coefficients between hyperparameters and model performance when the Rotation Trick is enabled.

| Hyperparameter | 6 | 8 | 16 | 32 | 64 | 128 | 256 | Avg. |
|---|---|---|---|---|---|---|---|---|
| Commitment Loss Weight | -0.67 | -0.82 | -0.93 | -0.84 | -0.69 | -0.50 | -0.53 | -0.71 |
| Codebook Size | 0.73 | 0.20 | 0.15 | 0.55 | 0.18 | 0.77 | 0.53 | 0.45 |
| Code Restart Threshold | -0.23 | -0.43 | -0.22 | -0.07 | 0.04 | -0.20 | -0.03 | -0.14 |
| EMA Decay | 0.13 | 0.21 | -0.03 | 0.17 | -0.03 | -0.06 | 0.12 | 0.06 |

Table 9: Correlation coefficients between hyperparameters and model performance when the Rotation Trick is disabled.

| Hyperparameter | 6 | 8 | 16 | 32 | 64 | 128 | 256 | Avg. |
|---|---|---|---|---|---|---|---|---|
| Commitment Loss Weight | 0.14 | -0.44 | -0.79 | -0.69 | -0.75 | -0.56 | -0.73 | -0.57 |
| Codebook Size | 0.69 | 0.17 | 0.29 | 0.48 | 0.38 | 0.76 | 0.81 | 0.52 |
| Code Restart Threshold | 0.13 | 0.33 | 0.05 | 0.08 | -0.01 | -0.21 | -0.15 | 0.02 |
| EMA Decay | -0.17 | 0.39 | 0.06 | -0.11 | -0.05 | 0.24 | 0.22 | 0.08 |

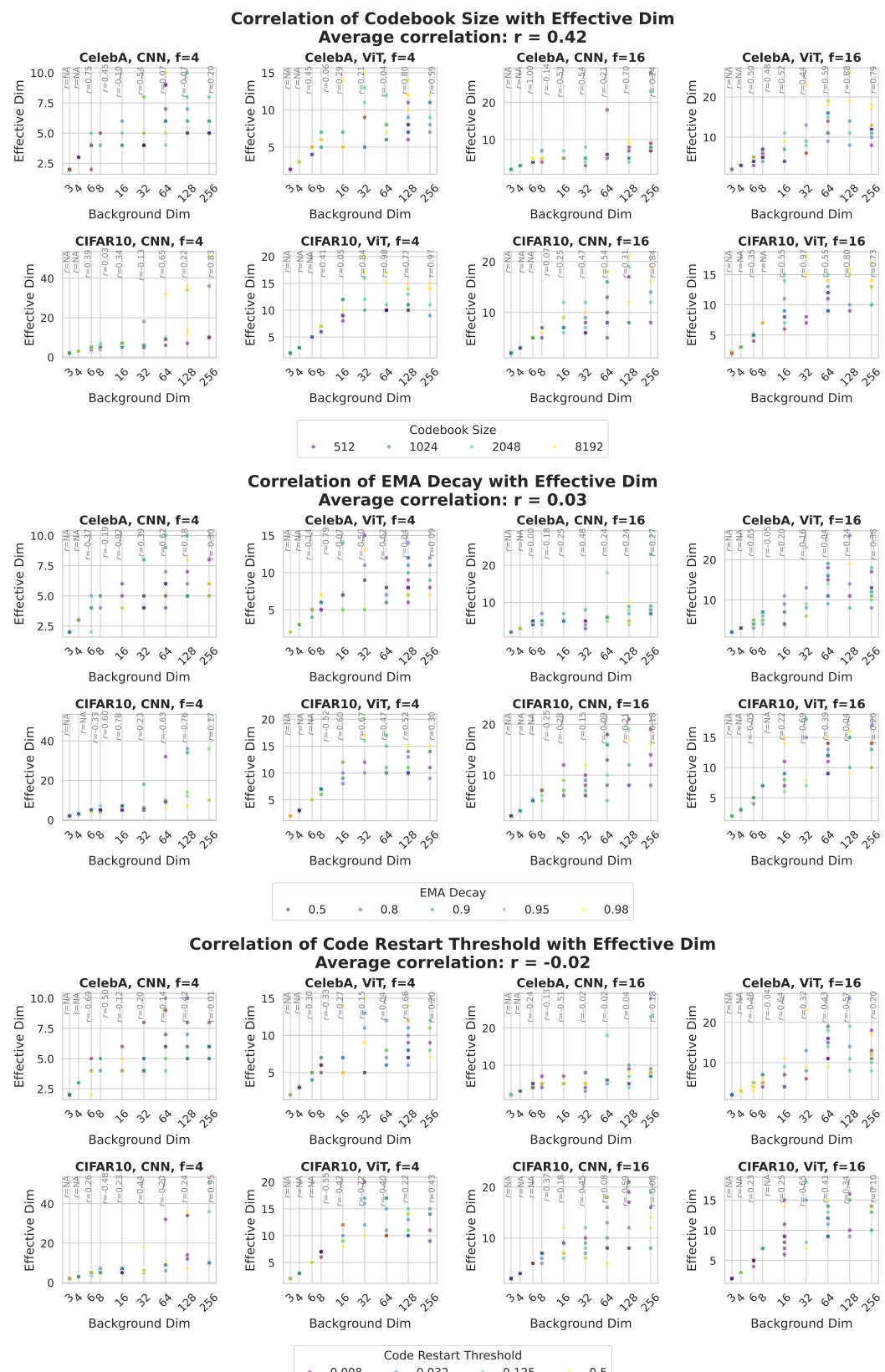

Figure 12: Correlation between effective dimensionality and various hyperparameters (part 1).

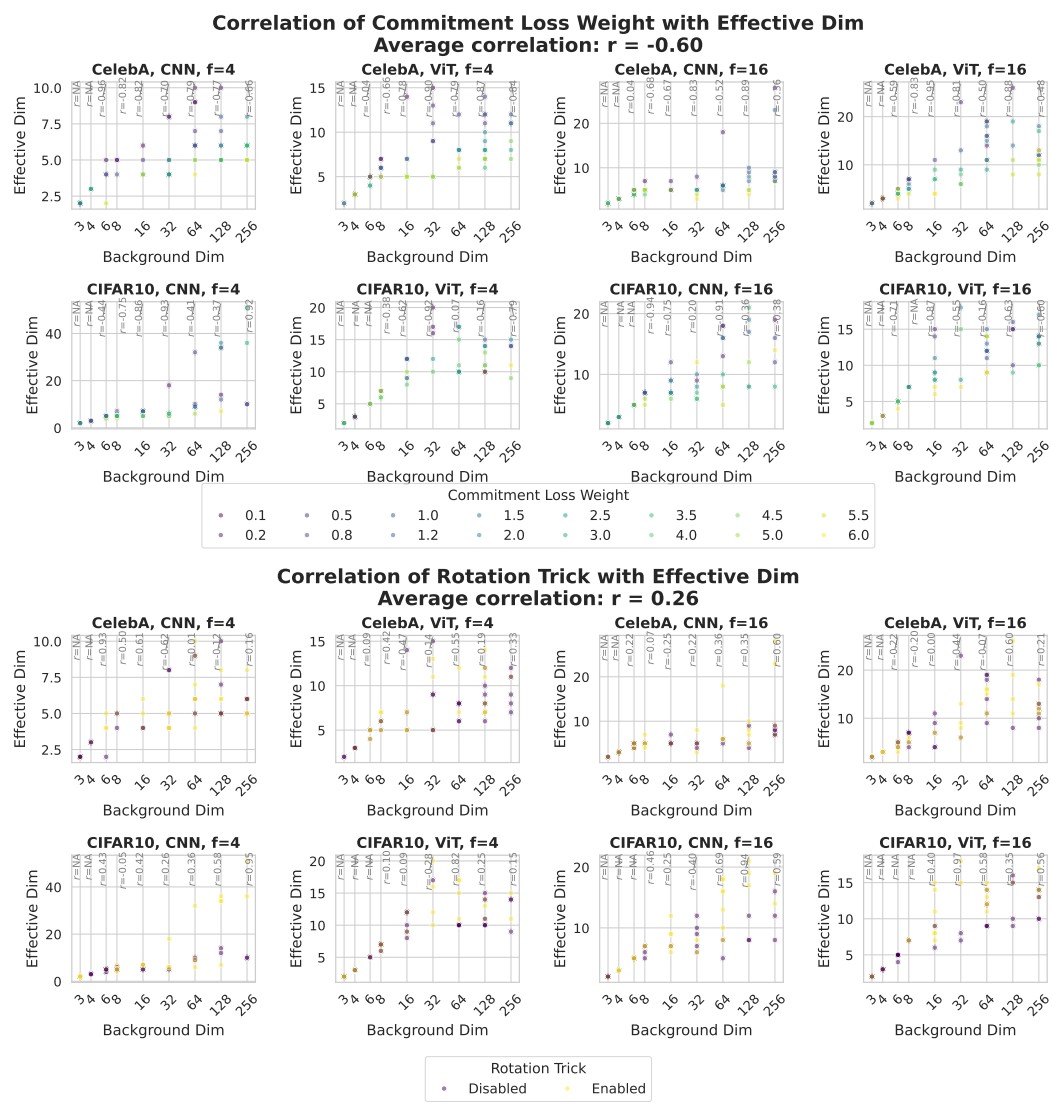

Figure 12 (continued): Additional correlation plots.

# F   Computational cost of DCVQ

To assess the computational overhead of DCVQ, we compare its training time with that of a standard VQVAE under identical conditions. All models are trained on CIFAR-10 using the same CNN encoder–decoder architecture for 100 epochs on a single NVIDIA A100 GPU. In DCVQ, each subspace quantizer operates independently, allowing for efficient parallel processing. The results are summarized in Table 10.

Table 10: Comparison of training time between DCVQ and vanilla VQVAE on CIFAR-10. The total latent dimensionality is matched in each pair of experiments.

| Total dim | Type | Dim per quantizer | Num. quantizers | Training time |
| --- | --- | --- | --- | --- |
| 32 | VQVAE | – | – | 5h 34m 41s |
| 32 | DCVQ | 8 | 4 | 5h 34m 35s |
| 64 | VQVAE | – | – | 5h 40m 41s |
| 64 | DCVQ | 8 | 8 | 5h 40m 17s |
| 256 | VQVAE | – | – | 6h 17m 57s |
| 256 | DCVQ | 32 | 8 | 6h 13m 39s |

The training times of DCVQ and vanilla VQVAE are nearly identical across all settings when the total latent dimensionality is matched. This demonstrates that the additional quantization operations introduced by DCVQ incur negligible computational overhead.

# G    Synthetic Experiment Demonstrating Quantization Bias

The following PyTorch code reproduces the synthetic $k$-means experiment shown in Figure 7.

```python
import torch
import matplotlib.pyplot as plt
from sklearn.cluster import KMeans

def gen_data(N, D, p):
    eigvals = (10 ** torch.randn(D)).pow(p).sort(descending=True)[0]
    U, _, Vt = torch.linalg.svd(torch.randn(N, D), full_matrices=False
        )
    X = U @ torch.diag(torch.sqrt((N - 1) * eigvals)) @ Vt
    return X, eigvals

def centroid_eigvals(X, n_clusters=512):
    kmeans = KMeans(n_clusters=n_clusters, random_state=0, n_init=10)
    kmeans.fit(X.numpy())
    cov = torch.cov(torch.tensor(kmeans.cluster_centers_).T)
    return torch.tensor(kmeans.cluster_centers_), torch.linalg.svd(cov
        )[1]

torch.manual_seed(0)
N, D, K = 4096, 64, 512
X, target_eigs = gen_data(N, D, p=0.3)
centroids, centroid_eigs = centroid_eigvals(X, n_clusters=K)
ratio = centroid_eigs / target_eigs

fig, axs = plt.subplots(1, 2, figsize=(10, 4))

axs[0].plot(ratio.numpy(), 'o-', color='tab:blue', markersize=4)

U, S, Vt = torch.linalg.svd(X, full_matrices=False)
X_2d = X @ Vt[[0, -1], :].T
C_2d = centroids @ Vt[[0, -1], :].T
axs[1].scatter(X_2d[:, 0], X_2d[:, 1], s=5, alpha=0.3, color='tab:gray
    ', label='Data')
axs[1].scatter(C_2d[:, 0], C_2d[:, 1], s=30, color='tab:red', label='
    Centroids')
plt.show()
```

Code 1: Synthetic experiment showing eigenvalue suppression under $k$-means quantization.

