# OpenReview forum: "Dimensional Collapse in VQVAEs: Evidence and Remedies"
_NeurIPS.cc/2025/Conference — NeurIPS 2025 poster_

### Official Review · Reviewer_JcVR · 2025-06-01

**Clarity:** 3
**Significance:** 3
**Originality:** 3
**Rating:** 4
**Confidence:** 4

**Summary:**

This paper investigates a consistent phenomenon known as "dimensional collapse" in VQVAEs, where high-dimensional embeddings are effectively compressed into a much lower-dimensional subspace in practice. The authors provide a thorough empirical analysis of this phenomenon across various backbones and modalities, exploring its implications for model performance and learning dynamics. To address the limitations imposed by low-dimensional usage, the paper proposes Divide-and-Conquer VQ (DCVQ), which partitions the latent space into independently quantized low-dimensional subspaces, thereby expanding overall model capacity. Experimental results demonstrate that DCVQ overcomes the dimensional bottleneck and improves reconstruction quality on several image datasets.

**Questions:**

1. Why is there no dedicated Related Work section? How does this work relate to previous studies on dimensional collapse in self-supervised and representation learning?
2. Have the authors considered comparing DCVQ with other methods that address dimensional collapse, such as those based on feature covariance regularization or network weight constraints?
3. Can the authors provide experimental results on larger datasets to demonstrate the scalability and robustness of the proposed approach?
4. What is the computational overhead introduced by DCVQ’s divide-and-conquer scheme, especially in terms of quantization time? Are there trade-offs in efficiency?

**Ethical Concerns:**

["NO or VERY MINOR ethics concerns only"]

**Final Justification:**

The new experiments have addressed my primary concerns

**Limitations:**

1. The lack of engagement with prior literature on dimensional collapse and related phenomena is a major limitation, as it reduces the paper's impact and novelty from a broader perspective.
2. Methodological comparisons are insufficient. The effectiveness and efficiency of DCVQ would be better validated with comparisons against a wider range of established techniques.
3. The experimental scope is limited to relatively small datasets, raising concerns about the generality and practicality of the proposed approach for real-world, large-scale tasks.
4. The time or computational overhead incurred by the DCVQ method is not discussed or measured, which is important for assessing its applicability in resource-constrained scenarios.

**Quality:**

2

**Strengths And Weaknesses:**

Strengths:

1. The authors conduct a comprehensive empirical study of the dimensional collapse phenomenon in VQVAE models, covering multiple architectures and datasets. This extensive analysis helps raise awareness of an often-overlooked property of VQ-based models.
2. The observation that effective rank and model performance are not strictly correlated is novel and merits further investigation.
3. The proposed DCVQ method is intuitive and directly addresses the identified limitation, showing tangible improvements in reconstruction accuracy.

Weaknesses:

1. The paper lacks a Related Work section, and overall citation coverage is insufficient. The phenomenon of dimensional collapse has been studied extensively in representation learning and self-supervised learning literature. The omission of this context makes it difficult to position the paper relative to prior research.
2. The comparative analysis is limited. While the paper uses rank regularization to encourage higher-dimensional usage, other established approaches, such as network weight constraints and feature covariance regularization, are not discussed or empirically compared.
3. The experimental evaluation is restricted to relatively small datasets. It would significantly strengthen the work to include results on larger-scale datasets such as ImageNet-100.
4. The paper does not provide a detailed assessment of the computational cost introduced by the DCVQ approach, particularly regarding the time overhead from multiple quantizations.

---

> ### Author Rebuttal · Authors · 2025-07-31
>
> # Response to Reviewer JcVR
> We greatly appreciate the reviewer’s thorough evaluation and insightful feedback. Your comments have been invaluable in helping us improve the clarity and rigor of our work. We address each of your points in detail below.
>
> ## Response to Q1
>
> We thank the reviewer for this important comment. Due to page limits, we placed the Related Work section in the appendix, but we will make this clearer in the revised version. We also appreciate the observation that “dimensional collapse has been studied extensively in representation learning and self-supervised learning literature.” Accordingly, we now include more regularizers inspired by self-supervised methods (see our later response).
>
> We would like to clarify, however, that the collapse we study in VQVAEs differs from that in self-supervised settings: while the representational collapse of self-supervised arises from forcing invariant output, the collapse in VQVAEs is due to the quantization mechanism, which are inherently different.
>
> ## Response to Q2
>
> We thank the reviewer for pointing out the limited scope of the comparative analysis. To address this, we extended our study beyond rank regularization to include several established approaches, such as Barlow Twins and VICReg from the self-supervised learning literature. Though originally designed for contrasting augmented views, these methods can be adapted to our setting for avoiding representational collapse. We also present a custom regularizer (SingHinge) designed to encourage singular values of the feature covariance matrix to be above a certain threshold. Detailed implementations of these regularizers are provided later in this response.
>
>
> **Table 1: CIFAR10 performance with different regularizers**
>
> | Regularizer Type|   Regularizer Weight |   Val Loss  |   Effective Dim  |
> |-----------------|--------------|-------------|------------------|
> | None|  -   |   0.081 |9 |
> | KoLeo   |   0.001  |   0.081 |9 |
> | KoLeo   |   0.01   |   0.082 |   10 |
> | KoLeo   |   0.1|   0.106 |   27 |
> | BarlowTwins |   0.0001 |   0.081 |   10 |
> | BarlowTwins |   0.001  |   0.093 |   31 |
> | BarlowTwins |   0.01   |   0.105 |   78 |
> | BarlowTwins |   0.1|   0.087 |   49 |
> | VIC |   0.1|   0.081 |9 |
> | VIC |   1  |   0.080 |   10 |
> | VIC |  10  |   0.082 |   11 |
> | VIC | 100  |   0.085 |   12 |
> | SingHinge(0.01) |   0.1|   0.080 |9 |
> | SingHinge(0.01) |   1  |   0.081 |9 |
> | SingHinge(0.01) |  10  |   0.089 |   49 |
> | SingHinge(0.01) | 100  |   0.129 |  111 |
> | SingHinge(0.1)  |   0.1|   0.080 |9 |
> | SingHinge(0.1)  |   1  |   0.081 |   10 |
> | SingHinge(0.1)  |  10  |   0.084 |   11 |
> | SingHinge(0.1)  | 100  |   0.085 |   12 |
>
> (All regularizers were applied on the unquantized latents. Due to the time limit, we only tested the CNN architecture with $f=4$. Val Loss reports validation reconstruction loss only, excluding the regularization term. SingHinge($h$) applies a hinge loss on singular values with threshold $h$ (e.g., 0.01 or 0.1). Regularization weights were chosen to yield performance comparable to the no-regularization baseline.)
>
> From these results, we observe a consistent pattern: stronger regularization increases the effective dimensionality but also leads to performance degradation once the dimension grows beyond the baseline level (≈9). Importantly, this trend appears robust across diverse regularization strategies, suggesting that the limitation is not specific to a single method or optimization artifact, but instead reflects a fundamental property of the VQ quantization bottleneck.
>
> ## Response to Q3
>
> We appreciate the reviewer’s suggestion to strengthen our evaluation with larger-scale datasets. We would like to clarify that Section 5.2 already includes ImageNet-1k experiments for our method and RQVAE. To further address the reviewer’s concern, we extended the evaluation to include vanilla VQVAEs and additional regularizers on ImageNet-1k:
>
> **Table 2: ImageNet-1k performance with different methods**
> | Type|   Regularizer Weight |   Val Loss  |   Effective Dim  |
> |-----------------|--------------|-------------|------------------|
> | BarlowTwins |   0.0001 |   0.173 |   14 |
> | BarlowTwins |   0.001  |   0.186 |   41 |
> | BarlowTwins |   0.01   |   0.205 |   91 |
> | BarlowTwins |   0.1|   0.182 |   58 |
> | VIC |   0.1|   0.172 |9 |
> | VIC |   1  |   0.164 |   11 |
> | VIC |  10  |   0.169 |   13 |
> | VIC | 100  |   0.18  |   20 |
> | SingHinge(0.01) |   0.1|   0.158 |9 |
> | SingHinge(0.01) |   1  |   0.168 |   11 |
> | SingHinge(0.01) |  10  |   0.181 |   63 |
> | SingHinge(0.01) | 100  |   0.247 |  137 |
> | SingHinge(0.1)  |   1  |   0.171 |   11 |
> | SingHinge(0.1)  |  10  |   0.183 |   15 |
> | SingHinge(0.1)  | 100  |   0.186 |   17 |
> | DCVQ(4x8d)  |   - |   0.115 |   20 |
> | DCVQ(8x8d)  |   - |   0.105 |   36 |
>
> Consistent with our CIFAR10 findings, we observe that increasing the strength of Barlow Twins, VICReg, or SingHinge raises the effective dimensionality but eventually degrades validation performance once dimensionality exceeds the baseline regime. This trend suggests that the trade-off between dimensional usage and reconstruction quality is robust across both small-scale and larger-scale datasets.
>
> Please note that, due to time constraints, the results presented here are from training runs stopped early (21k steps), before full convergence. At this stage, the effective dimensions had plateaued, making it possible to compare the influence of regularizers. Differences from Section 5.2 also include the use of smaller parameter sizes. We plan to provide full-convergence results in the camera-ready version.
>
> ## Response to Q4
>
> We thank the reviewer for raising the important point about the computational cost of DCVQ. We emphasize that the computational overhead of DCVQ is minimal, as the processing of each subspace is parallelized. To quantify this, we report the actual training times of DCVQ compared to vanilla VQVAE on CIFAR10 using a CNN architecture with $f=16$, trained for 100 epochs on an A100 GPU:
>
> | Total dim | Type        | Dim per quantizer  | Num quantizers | Training time |
> |-----------|-------------|--------------------|----------------|---------------|
> | 32        | VQVAE       | -                  | -              | 5h 34m 41s    |
> | 32        | DCVQ        | 8                  | 4              | 5h 34m 35s    |
> | 64        | VQVAE       | -                  | -              | 5h 40m 41s    |
> | 64        | DCVQ        | 8                  | 8              | 5h 40m 17s    |
> | 256       | VQVAE       | -                  | -              | 6h 17m 57s    |
> | 256       | DCVQ        | 32                 | 8              | 6h 13m 39s    |
>
> From these results, we observe that the training time of DCVQ is essentially the same as that of vanilla VQVAE when the total dimensionality is matched. This indicates that the additional quantizations in DCVQ introduce negligible overhead.
>
>
> ## Implementation Details for Regularizers
>
> For **Barlow Twins** and **VICReg**, which were originally designed for contrastive learning with augmented view pairs, data augmentation is not needed in our rank-promotion setting. Instead, we form input pairs by duplicating the same latent vector. All other hyperparameters follow the original codebase.
>
> ### Barlow Twins Loss
>
> Given normalized latent vectors $ x, y \in \mathbb{R}^{B \times D} $, where $B$ denotes the total number of patches across GPUs and $D$ the feature dimension, and letting $\text{BN}$ denote 1D batch normalization, the Barlow Twins loss is defined as:
>
> $$
> C = \frac{1}{B} \, \text{BN}(x)^\top \text{BN}(y) \\\\
> \mathcal{L}_{\text{Barlow}} = \sum_i (C_{ii} - 1)^2 + 0.0051 \sum_{i \neq j} C_{ij}^2
> $$
>
> ### VICReg Loss
>
> Given $x, y, B, D$ defined similarly, the VICReg loss combines variance and covariance regularization:
>
> The variance term:
>
> $$
> \mathcal{L}_{\text{var}}
> = \frac{1}{D} \sum_{i=1}^D \frac{\max(0, 1 - \sigma(x_{\cdot i})) + \max(0, 1 - \sigma(y_{\cdot i}))}{2}
> $$
>
> where $\sigma$ denotes the standard deviation.
>
> The covariance term:
>
> $$
> \mathcal{L}_{\text{cov}} = \frac{1}{D} \sum_{i \neq j} (\text{Cov}(x)_{ij}^2 + \text{Cov}(y)_{ij}^2)
> $$
>
> with $\text{Cov}(x) = \tfrac{1}{B-1}(x - \mu(x))^\top(x - \mu(x))$, where $\mu(x)$ is the batch mean.
>
> The total VICReg loss is then:
>
> $$
> \mathcal{L}_{\text{VICReg}} = \mathcal{L}_{\text{var}} + \tfrac{1}{25} \cdot \mathcal{L}_{\text{cov}}
> $$
>
> ### SingHinge
>
> For latent representations $z \in \mathbb{R}^{B \times D}$, let $s_i$ denote the singular values of $\text{Cov}(z)$. With a threshold $h$, the SingHinge loss is defined as
>
> $$
> \mathcal{L}_{\text{SingHinge}(h)} = \frac{1}{D} \sum_i \max(0, h - s_i)
> $$
>
> **References:**
>
> [1] Barlow Twins: Self-Supervised Learning via Redundancy Reduction
>
> [2] VICReg: Variance-Invariance-Covariance Regularization for Self-Supervised Learning

---

> > ### Comment · Reviewer_JcVR · 2025-08-05
> >
> > Thank you for the detailed response during the rebuttal period. The new experiments have addressed my primary concerns.
> > I encourage the authors to incorporate these experiments and the corresponding discussion into the final version of the paper.

---

> > > ### Author Response · Authors · 2025-08-05
> > >
> > > Thank you for your thoughtful feedback. We’re pleased to know that our additional experiments have addressed your concerns. We will incorporate these results along with the corresponding discussion into the final version of the paper.

---

### Official Review · Reviewer_21RK · 2025-06-30

**Clarity:** 3
**Significance:** 3
**Originality:** 3
**Rating:** 4
**Confidence:** 5

**Summary:**

The paper uncovers an interesting phenomenon where VQ-VAEs tend to compress their representations into a much smaller subspace, despite being trained with high-dimensional embeddings. Through a thorough analysis, the authors demonstrate how this compression relates to model performance and learning dynamics. To address the limitations imposed by this low-dimensional bottleneck, they propose the Divide-and-Conquer VQ (DCVQ) method, which partitions the latent space into multiple low-dimensional subspaces that are quantized independently. The combined effect of these subspaces significantly expands the overall representational capacity. Experimental results show that DCVQ effectively overcomes the dimensionality bottleneck and achieves improved reconstruction quality across various image datasets.

**Questions:**

It is necessary to include comparisons with VQGAN-FC (which performs dimensionality reduction and normalization), while also controlling for the same amount of information bits in the codebooks.

**Ethical Concerns:**

["NO or VERY MINOR ethics concerns only"]

**Final Justification:**

This paper raises an important and novel issue but is not technically solid.

**Limitations:**

yes

**Quality:**

3

**Strengths And Weaknesses:**

### Strengths
- The identification of the dimension collapse problem in VQ-VAEs is insightful and worthy of further exploration.
- The experiments and analyses are comprehensive and well-executed.

### Weaknesses
- I had hoped the authors would uncover the fundamental cause of dimension collapse and provide a thorough explanation, possibly from the perspective of optimization dynamics. Unfortunately, the paper only observes that the quantizer collapses first, which then causes the encoder to collapse, without deeper theoretical insight.
- The proposed Divide-and-Conquer method still relies on low-dimensional codebook subspaces, making the approach fundamentally similar to previous dimensionality reduction techniques (e.g., Improved VQGAN), lacking a truly novel solution.

---

> ### Author Rebuttal · Authors · 2025-07-31
>
> # Response to Reviewer 21RK
>
> We sincerely thank the reviewer for the thoughtful review and for the positive assessment of our work. We greatly appreciate your constructive feedback, and we have carefully considered each of your comments in detail. Below we provide clarifications and additional context.
>
> ## Response to "I had hoped the authors would uncover the fundamental cause of dimension collapse"
>
> We appreciate this insightful question, as understanding the fundamental cause of dimension collapse is indeed central to advancing this line of work. While a full theoretical characterization remains an open challenge, our experiments provide evidence for a plausible underlying mechanism.
>
> As discussed in the paper, the quantizer is the first module to collapse. We attribute this not to incidental factors but to an inherent bias of the quantization process. Specifically, VQ‑VAE’s quantization can be viewed as an online K‑Means procedure. A synthetic experiment can illustrate that K‑Means systematically reduces the effective dimension of the data: the learned centroids bias towards with the high‑variance directions of the data covariance matrix. Concretely, the ratio between centroid eigenvalues and data eigenvalues stays close to 1 for the largest eigenvalues but drops significantly for smaller ones. Thus, high‑variance components are well preserved, whereas low‑variance components are suppressed, leading to a reduced effective dimension in the codebook compared to the original data.
>
> The behavior could be illustrated by the snippet below. The experiment constructs synthetic data with a controlled covariance spectrum, applies K‑Means clustering, and compares the eigenvalue decay before and after quantization. The results consistently show that smaller eigenvalues shrink more severely, confirming the bias towards high‑variance directions.
>
> Our findings are also supported by prior theoretical work linking K‑Means and PCA, including [1] Spectral Relaxation for K‑Means Clustering and [2] K‑Means Clustering via Principal Component Analysis. These works suggest that clustering inherently emphasizes dominant principal components, providing a theoretical lens through which to interpret the observed rank vanishing.
>
> While a rigorous proof remains an exciting direction for future research, we believe that the empirical evidence presented here, together with prior literature, strongly indicates that dimension collapse arises as a structural property of quantization.
>
> ```python
> import torch
> import matplotlib.pyplot as plt
> from sklearn.cluster import KMeans
>
> def gen_data(N, D, p):
>     eigvals = (10 ** torch.randn(D)).pow(p).sort(descending=True)[0]
>     U, _, Vt = torch.linalg.svd(torch.randn(N, D), full_matrices=False)
>     X = U @ torch.diag(torch.sqrt((N - 1) * eigvals)) @ Vt
>     return X, eigvals
>
> def centroid_eigvals(X, n_clusters=512):
>     kmeans = KMeans(n_clusters=n_clusters, random_state=0, n_init=10)
>     kmeans.fit(X.numpy())
>     cov = torch.cov(torch.tensor(kmeans.cluster_centers_).T)
>     return torch.linalg.svd(cov)[1]
>
> torch.manual_seed(0)
>
> N, D, K = 4096, 64, 512
> X, target_eigs = gen_data(N, D, p=0.3)
> centroid_eigs = centroid_eigvals(X, n_clusters=K)
> ratio = centroid_eigs / target_eigs
>
> plt.plot(ratio.numpy(), 'o-')
> plt.axhline(1.0)
> plt.xlabel('Eigenvalue Index (Largest to Smallest)')
> plt.ylabel('Centroid / Data Eigenvalue')
> plt.yscale('log')
> plt.show()
> ```
>
> ## Response to "The proposed Divide-and-Conquer method still relies on low-dimensional codebook subspaces ... lacking a truly novel solution."
>
> We thank the reviewer for this comment. We would like to clarify that Improved VQGAN does not perform dimensionality reduction; instead, it aggregates high-dimensional codebook vectors, which unfortunately still leads to collapsed representations. Our proposed DCVQ is explicitly designed to mitigate this issue, enabling more effective utilization of the latent space.
>
> We also wish to emphasize that, beyond proposing a method, our work makes a contribution by identifying and systematically analyzing the collapse phenomenon itself — an aspect that, to our knowledge, has not been studied in depth in prior literature.
>
>
> ## Repsonse to Q1: "It is necessary to include comparisons with VQGAN-FC ..., while also controlling for the same amount of information bits in the codebooks."
>
> We greatly appreciate this suggestion. While we did not explicitly label it as “VQGAN‑FC,” we believe the requested comparison is already covered in our experiments. Specifically, in §3.2.1 we introduced a projection layer after the encoder and a corresponding inverse projection before the decoder to vary the background dimension while keeping the encoder architecture unchanged. This setup is equivalent to the Factorized Codes approach.
>
> In Table 1, we report results across a wide range of background dimensions (3, 4, 6, 8, 16, 32, 64, 128, 256), directly testing different levels of factorization. Furthermore, Figure 5 shows how varying the background dimension affects the effective dimensionality, thereby addressing the reviewer’s concern.
>
> Regarding normalization, we note that all models in our experiments apply L2 normalization before quantization (as described in §3.2.1). The only exception is in §5, where Residual VQ does not support L2 normalization; in this case, we disabled it for our method as well to ensure fairness.
>
> Finally, we ensured comparability by controlling the total information content: the number of tokens and codebook sizes were kept consistent across methods, ensuring that the effective information capacity was comparable.
>
> **References:**
>
> [1] Spectral Relaxation for K-means Clustering
>
> [2] K-means Clustering via Principal Component Analysis

---

> > ### Comment · Reviewer_21RK · 2025-08-04
> >
> > After reading the author's response and the comments from other reviewers, I agree that this paper raises an important and novel issue regarding the dimensional collapse problem in VQ. However, similar to the concerns expressed by the other reviewers, I remain significantly concerned about the proposed solution, particularly its reliance on explicit dimensionality reduction and regularization techniques. Therefore, I consider this a borderline paper and will maintain my original score.

---

> > > ### Author Response · Authors · 2025-08-05
> > >
> > > Thank you for acknowledging the importance of the dimensional collapse issue in VQ. We understand your concerns and appreciate your constructive feedback. We will carefully consider your points to further strengthen our approach and clarify its contributions in the final version.

---

### Official Review · Reviewer_TLST · 2025-07-01

**Clarity:** 2
**Significance:** 4
**Originality:** 3
**Rating:** 5
**Confidence:** 3

**Summary:**

This paper identifies a previously unrecognized issue in vector quantized variational autoencoders (VQVAEs), termed dimensional collapse, wherein the learned codebooks tend to collapse onto a limited number of principal components, even when the dimensionality of the code vectors is large. The authors support their findings through extensive experiments, showing that the phenomenon arises from the quantization step during the early training phase. To address this issue, the latent representation is divided into smaller vectors, each of which is independently quantized. This technique enhances both the effective dimensionality of the codebooks and the overall reconstruction quality.

**Questions:**

1. The detailed explanation of the projection layer following the encoder may help clarify the architectures. Is this layer also trained during the training?
2. In Figure 9, the compared models employ multiple codebooks. Could the authors clarify how the effective dimension was computed in this multi-codebook setting?
3. In the experimental results shown in Figures 4 and 5, the effective dimension remained below 20 even when hyperparameters such as $f$ and background dimension were varied. However, in Figure 9, effective dimensions around 100 were reported. Could the authors briefly explain which aspect of the proposed method leads to this discrepancy?
4. While I agree that training dynamics of the quantizers may cause dimension collapse, I am curious about the authors’ thoughts on why this phenomenon occurs. Since the codebook is updated as cluster centers of the encoding vectors, implying that encoding vectors are mapped near a low-dimensional subspace, any hypothesized reasoning behind this behavior would be very interesting.
5. I am also curious about the potential relationship between the effective dimension and the intrinsic dimension of the data. Could the authors comment on this?

**Ethical Concerns:**

["NO or VERY MINOR ethics concerns only"]

**Final Justification:**

The authors have addressed most of my initial concerns and provided compelling empirical evidence relevant to my question. The identified issue in this paper has the potential to be influential in this field, and its validity is supported by well-designed experiments.

**Limitations:**

yes

**Paper Formatting Concerns:**

I did not find any issues related to the paper formatting.

**Quality:**

3

**Strengths And Weaknesses:**

### Strengths
1. The writing is well-written, with clear and easy-to-follow logical flow.
2. The authors effectively design experiments that validate their hypotheses, and their results provide clear evidence, supported by sound interpretation.
3. The authors propose a simple yet effective method to address the identified issue.

### Weaknesses
1. The encoder scale factor $f$ controls the number of encoding vectors and likely influences the expressive capacity of the latent representation. It may also be related to the effective dimension of the model. Although the experiments include comparisons with $f=4$ and $f=16$, the main text does not provide a detailed interpretation of how this parameter affects the results. Including such an analysis would help clarify the role of $f$ and its relationship to the model's representational capacity.
2. The proposed method shows improvements in reconstruction quality, but does not discuss the generation procedure or the quality of generated samples. Including such analysis would help assess the model as a generative model.
3. I would like to offer a few comments regarding the figures and tables, which may help improve the clarity and presentation of the results.
    1. In Figure 4, the size of the circles is not clearly distinguishable, which makes it difficult to interpret the relationships among background dimension, effective dimension, and validation loss.
    2. In Figure 5, using a logarithmic scale on the x-axis may improve readability. Also, it would be helpful to clarify what the colors represent.
    3. The notation $d_{\text{eff}}$​ is defined in the text, but it does not appear in the figures or results. Referring to it explicitly in the relevant graphs would help improve clarity. Similarly, in Table 1, it would be helpful to indicate how each hyperparameter corresponds to components in the formulation presented in Section 2.1.
    4. In Figure 7, the legend or caption does not explain the meaning of the different line styles (e.g., circle line for validation loss, square line for effective dimension). Providing this information would help readers interpret the figure correctly.
    5. For completeness and consistency, it may be useful to test the model under the case $f = 4$ as well.
    6. In Figure 9, it is unclear which parameter(s) of RQVAE were varied to generate the graph, as this is not described in the main text. Additionally, since DCVQ can be interpreted as increasing the codebook size and decreasing $f$ (which corresponds to increasing the number of encoding vectors), it would be beneficial to include the standard RQVAE with these settings as a baseline for comparison.
    7. In Table 2, reporting regression coefficients instead of correlations may better reveal the influence of each latent variable. For interpretability, it may also be helpful to exclude the binary variable associated with the rotation trick when fitting the regression model.
4. Some additions to the manuscript could improve clarity and completeness.
    1. Table 1 includes terms such as “rotation trick” and “dead-code restart threshold,” but these are not explained in the main text. Providing brief explanations would help make the paper more self-contained.
    2. Including training details such as the optimizer, learning rate, batch size, and number of epochs (perhaps in an appendix) would help improve the reproducibility of the experiments.

---

> ### Author Rebuttal · Authors · 2025-07-31
>
> # Response to Reviewer TLST
>
> We are truly grateful to the reviewer for taking the time to carefully read our work and provide such constructive suggestions. Your feedback has been instrumental in refining our manuscript. Below, we respond to each of your comments and questions.
>
> ## Response to Weakness 1
>
> We omitted detailed comparisons earlier because $f$ strongly affects validation loss, making figures hard to interpret. Table 1 now reports the values from Figure 7, grouped by scale factor $f$. Results show that effective dimension is not systematically influenced by $f$, while validation loss is. Under KoLeo, effective dimension varies with $f$, but without a consistent trend—likely due to regularization-induced instability.
>
> **Table 1. Comparison of different $f$**
> | Dataset   | Architecture   |   Val Loss |   Effective Dim |   KoLeo loss weight |
> |:----------|:---------------|-----------:|----------------:|----------:|
> | CIFAR10   | CNN(f=16)  |  0.480 |   8 | 0 |
> | CIFAR10   | CNN(f=4)   |  0.081 |   9 | 0 |
> | CIFAR10   | ViT(f=16)  |  0.480 |   9 | 0 |
> | CIFAR10   | ViT(f=4)   |  0.084 |   9 | 0 |
> | CIFAR10   | CNN(f=16)  |  0.482 |   9 | 0.001 |
> | CIFAR10   | CNN(f=4)   |  0.081 |   9 | 0.001 |
> | CIFAR10   | ViT(f=16)  |  0.479 |   9 | 0.001 |
> | CIFAR10   | ViT(f=4)   |  0.084 |   9 | 0.001 |
> | CIFAR10   | CNN(f=16)  |  0.492 |  10 | 0.01  |
> | CIFAR10   | CNN(f=4)   |  0.082 |  10 | 0.01  |
> | CIFAR10   | ViT(f=16)  |  0.480 |   9 | 0.01  |
> | CIFAR10   | ViT(f=4)   |  0.084 |   9 | 0.01  |
> | CIFAR10   | CNN(f=16)  |  0.523 |  15 | 0.1   |
> | CIFAR10   | CNN(f=4)   |  0.106 |  27 | 0.1   |
> | CIFAR10   | ViT(f=16)  |  0.494 |  11 | 0.1   |
> | CIFAR10   | ViT(f=4)   |  0.090 |  12 | 0.1   |
> | CelebA| CNN(f=16)  |  0.033 |   7 | 0 |
> | CelebA| CNN(f=4)   |  0.003 |   7 | 0 |
> | CelebA| ViT(f=16)  |  0.034 |  10 | 0 |
> | CelebA| ViT(f=4)   |  0.004 |   9 | 0 |
> | CelebA| CNN(f=16)  |  0.033 |   7 | 0.001 |
> | CelebA| CNN(f=4)   |  0.003 |   7 | 0.001 |
> | CelebA| ViT(f=16)  |  0.034 |  11 | 0.001 |
> | CelebA| ViT(f=4)   |  0.004 |  10 | 0.001 |
> | CelebA| CNN(f=16)  |  0.034 |   9 | 0.01  |
> | CelebA| CNN(f=4)   |  0.003 |   8 | 0.01  |
> | CelebA| ViT(f=16)  |  0.036 |  14 | 0.01  |
> | CelebA| ViT(f=4)   |  0.004 |  11 | 0.01  |
> | CelebA| CNN(f=16)  |  0.043 |  41 | 0.1   |
> | CelebA| CNN(f=4)   |  0.005 |  84 | 0.1   |
> | CelebA| ViT(f=16)  |  0.038 |  60 | 0.1   |
> | CelebA| ViT(f=4)   |  0.007 |  37 | 0.1   |
>
> ## Response to Weakness 2
>
> We thank the reviewer for this valuable comment. We agree that evaluating the generation procedure and sample quality is essential. These experiments are ongoing, and we plan to include results soon.
>
> ## Response to Weakness 3
>
> We thank the reviewer for the detailed feedback on our figures and tables.
> 1. Fig. 4: We will improve circle size visibility in the revision.
> 2. Fig. 5: All points are plotted with the same color; the perceived differences arise from overlapping points. We will revise the plotting style to eliminate this artifact.
> 3. Notation / Table 1: We apologize for the oversight. The effective dimension $d_eff$ was denoted as Effective Dim in the figures. We will revise 2.1 to make it clearer.
> 4. Fig. 7: We will remove line styles, as colors suffice.
> 5. $f=4$ (Fig. 9): Omitted due to space, but we will add it.
> 6. Fig. 9 (10?) (RQVAE parameters): The varied parameter was the number of quantizers; we will revise the caption/description. We also plan to include the standard VQVAE as a baseline.
> 7. Table 2:
>  (a) We use correlations instead of regression coefficients since hyperparameters vary in scale (e.g., EMA decay vs. codebook size), making regression coefficients non‑comparable; correlations are scale‑independent.
>  (b) We agree the binary “Rotation Trick” should be excluded and will present correlations separately for enabled/disabled cases. Please check the updated Table 2.1 and Table 2.2. Conclusions remain unchanged.
>
> **Table 2.1 Correlation coefficients when Rotation Trick is enabled**
> | Hyperparameter | Background Dim   |   Correlation |
> |:-----------------------|:-----------------|--------------:|
> | Commitment Loss Weight | 6| -0.67 |
> | Commitment Loss Weight | 8| -0.82 |
> | Commitment Loss Weight | 16   | -0.93 |
> | Commitment Loss Weight | 32   | -0.84 |
> | Commitment Loss Weight | 64   | -0.69 |
> | Commitment Loss Weight | 128  | -0.50 |
> | Commitment Loss Weight | 256  | -0.53 |
> | Commitment Loss Weight | Average  | -0.71 |
> | Codebook Size  | 6|  0.73 |
> | Codebook Size  | 8|  0.20 |
> | Codebook Size  | 16   |  0.15 |
> | Codebook Size  | 32   |  0.55 |
> | Codebook Size  | 64   |  0.18 |
> | Codebook Size  | 128  |  0.77 |
> | Codebook Size  | 256  |  0.53 |
> | Codebook Size  | Average  |  0.45 |
> | Code Restart Threshold | 6| -0.23 |
> | Code Restart Threshold | 8| -0.43 |
> | Code Restart Threshold | 16   | -0.22 |
> | Code Restart Threshold | 32   | -0.07 |
> | Code Restart Threshold | 64   |  0.04 |
> | Code Restart Threshold | 128  | -0.20 |
> | Code Restart Threshold | 256  | -0.03 |
> | Code Restart Threshold | Average  | -0.14 |
> | EMA Decay  | 6|  0.13 |
> | EMA Decay  | 8|  0.21 |
> | EMA Decay  | 16   | -0.03 |
> | EMA Decay  | 32   |  0.17 |
> | EMA Decay  | 64   | -0.03 |
> | EMA Decay  | 128  | -0.06 |
> | EMA Decay  | 256  |  0.12 |
> | EMA Decay  | Average  |  0.06 |
>
> **Table 2.2 Correlation coefficients when Rotation Trick is disabled**
> | Hyperparameter | Background Dim   |   Correlation |
> |:-----------------------|:-----------------|--------------:|
> | Commitment Loss Weight | 6|  0.14 |
> | Commitment Loss Weight | 8| -0.44 |
> | Commitment Loss Weight | 16   | -0.79 |
> | Commitment Loss Weight | 32   | -0.69 |
> | Commitment Loss Weight | 64   | -0.75 |
> | Commitment Loss Weight | 128  | -0.56 |
> | Commitment Loss Weight | 256  | -0.73 |
> | Commitment Loss Weight | Average  | -0.57 |
> | Codebook Size  | 6|  0.69 |
> | Codebook Size  | 8|  0.17 |
> | Codebook Size  | 16   |  0.29 |
> | Codebook Size  | 32   |  0.48 |
> | Codebook Size  | 64   |  0.38 |
> | Codebook Size  | 128  |  0.76 |
> | Codebook Size  | 256  |  0.81 |
> | Codebook Size  | Average  |  0.52 |
> | Code Restart Threshold | 6|  0.13 |
> | Code Restart Threshold | 8|  0.33 |
> | Code Restart Threshold | 16   |  0.05 |
> | Code Restart Threshold | 32   |  0.08 |
> | Code Restart Threshold | 64   | -0.01 |
> | Code Restart Threshold | 128  | -0.21 |
> | Code Restart Threshold | 256  | -0.15 |
> | Code Restart Threshold | Average  |  0.02 |
> | EMA Decay  | 6| -0.17 |
> | EMA Decay  | 8|  0.39 |
> | EMA Decay  | 16   |  0.06 |
> | EMA Decay  | 32   | -0.11 |
> | EMA Decay  | 64   | -0.05 |
> | EMA Decay  | 128  |  0.24 |
> | EMA Decay  | 256  |  0.22 |
> | EMA Decay  | Average  |  0.08 |
>
> ## Response to Weakness 4
> 1. The definitions of the rotation trick and dead‑code restart threshold are provided in Related Work (Appendix A). We will expand on these explanations in the revision.
> 2. Training details are included in Appendix C.
>
> ## Response to Questions
> 1. The projection layer following the encoder is a trainable linear layer. This allows us to map the latents into arbitrary background dimensions. Similar tricks could be found in Improved VQGAN (RQVAE).
> 2. For RQVAE, the final representation is obtained by summing multiple codes, while the representational space is determined by the set of codes prior to summation. We concatenate all code vectors into a joint matrix and compute its effective dimension. In contrast, our method defines the space as a direct sum of subspaces, so its effective dimension equals the sum of the effective dimensions of the individual subspaces.
> 3. In Figure 9, effective dimensions of around 100 correspond to the DCVQ method. The figure illustrates how increasing the number of quantizers raises the effective dimension. For example, the notation $16 \times 8d$ indicates 16 quantizers, each with dimensionality $8d$.
> 4. Our hypothesis is that quantization biases towards high‑variance directions. Since quantization can be viewed as a variant of online K‑Means, we can illustrate this phenomenon in a controlled K‑Means experiment. Specifically, we generate synthetic data with a covariance matrix of desired eigenvalues, apply K‑Means clustering, and then examine the eigenvalues of the covariance of the cluster centroids. We find that the smaller eigenvalues shrink more severely after clustering, while the larger ones are preserved. As a result, the effective dimension of the centroids (i.e., the codebook) is reduced compared to that of the original data. The following snippet demonstrates this behavior. To develop a theoretical explanation for the rank‑vanishing behavior observed in K‑Means, one could connect the observations to existing frameworks, such as (1) Spectral Relaxation for K‑Means Clustering and (2) K‑Means Clustering via Principal Component Analysis. These studies suggest that K‑Means exhibits PCA‑like behavior, preferentially capturing the principal components of the data.
> 5. We assume that the effective dimension is more like an artifact of the quantization process, since no such collapse is observed in standard autoencoders.
>
> ```python
> import torch
> import matplotlib.pyplot as plt
> from sklearn.cluster import KMeans
>
> def gen_data(N, D, p):
>     eigvals = (10 ** torch.randn(D)).pow(p).sort(descending=True)[0]
>     U, _, Vt = torch.linalg.svd(torch.randn(N, D), full_matrices=False)
>     X = U @ torch.diag(torch.sqrt((N - 1) * eigvals)) @ Vt
>     return X, eigvals
>
> def centroid_eigvals(X, n_clusters=512):
>     kmeans = KMeans(n_clusters=n_clusters, random_state=0, n_init=10)
>     kmeans.fit(X.numpy())
>     cov = torch.cov(torch.tensor(kmeans.cluster_centers_).T)
>     return torch.linalg.svd(cov)[1]
>
> torch.manual_seed(0)
>
> N, D, K = 4096, 64, 512
> X, target_eigs = gen_data(N, D, p=0.3)
> centroid_eigs = centroid_eigvals(X, n_clusters=K)
> ratio = centroid_eigs / target_eigs
>
> plt.plot(ratio.numpy(), 'o-')
> plt.axhline(1.0)
> plt.xlabel('Eigenvalue Index (Largest to Smallest)')
> plt.ylabel('Centroid / Data Eigenvalue')
> plt.yscale('log')
> plt.show()
> ```

---

> ### Comment · Reviewer_TLST · 2025-08-04
>
> Thank you for your thoughtful and clarifying response. I appreciate that many of my concerns have been addressed, and I found your explanation regarding the quantization bias along the principal component direction particularly interesting. Highlighting this point in Section 3.3 would, in my view, add valuable depth to your discussion.
>
> While it is understandable that time constraints during the review period made it difficult to conduct additional generation experiments, I believe exploring the relationship between effective dimensionality and generation quality in future work could provide further important insights. With respect to Table 2, although correlation coefficients are certainly useful, I wonder if reporting standardized regression coefficients might better capture the individual effect of each variable while controlling for others, thus more directly serving your intended analysis.
>
> Lastly, the impact of $f$ (the degree of data compression) does not appear to be a central focus in the current manuscript. Given that one might intuitively expect a connection between $f$ and the effective dimension, I believe that explicitly discussing your empirical observation of no such relationship—and offering possible interpretations or hypotheses, as you did for the quantization bias—would further strengthen the contribution of your paper.
>
> Accordingly, if the authors adequately address the issues raised in this review—such as reorganizing figures and tables and providing thorough interpretations regarding quantization bias and the encoder scale factor $f$—I believe this paper merits a higher evaluation. For these reasons, I will raise my score to 5.

---

> > ### Author Response · Authors · 2025-08-05
> >
> > Thank you very much for your thoughtful and constructive feedback. We are glad our clarifications addressed many of your concerns, and we will incorporate your suggestions to further strengthen the paper. We sincerely appreciate your willingness to raise your score and your guidance in improving our work.

---

> ### Author Response · Authors · 2025-08-08
> **Gentle Reminder – Score Update and Mandatory Acknowledgement**
>
> Dear Reviewer TLST,
>
> I hope you are doing well.
>
> We are truly grateful for your constructive feedback and your kind note indicating that you would consider raising your score to 5 after our rebuttal. Your suggestions on clarifying the quantization bias discussion and other points have been invaluable to improving our paper.
>
> As the rebuttal period is coming to an end, we just wanted to kindly check whether you might have had the chance to update your score and complete the Mandatory Acknowledgement in the system. We fully understand everyone’s busy schedule, and we greatly appreciate the time and effort you’ve already dedicated to reviewing our work.
>
> Thank you again for your thoughtful engagement and support.
>
> Best regards,
>
> The authors

---

### Official Review · Reviewer_enVD · 2025-07-03

**Clarity:** 3
**Significance:** 3
**Originality:** 3
**Rating:** 4
**Confidence:** 3

**Summary:**

This paper revolves around VQ-VAE (and vector quantization overall), and centers around the claim that such models hardly use the full size of their code vectors: the linear dimension for such trained models is incredibly small compared to the feature dimension. When trying to regularize for higher rank features, performance drops off notably. The authors then offer a fix that applies to all VQ models: splitting into subspaces, then quantizing at each individually low-dimensional space. This keeps VQ in a low-dimensional regime while allowing for higher model capacity.

**Questions:**

1. Why does the KoLeo regularization give a strong statement regarding restriction to a higher rank set for optimization? That is, why are we not concerned that performance degredation could be due to the effect of this regularizer's gradient structure on optimization?
2. What other methods for rank maximization/thresholding would be viable to test against as well?
3. What are the visual effects of more/less subspaces, and higher/lower dimensions for each of the subspaces?

**Ethical Concerns:**

["NO or VERY MINOR ethics concerns only"]

**Final Justification:**

The main concern was around experimental verification of the rank claim, which is now much more robust. I have raised my score accordingly.

**Limitations:**

yes

**Quality:**

2

**Strengths And Weaknesses:**

The paper generally is very readable: the claim is easy to understand, and the experiments are controlled and interpretable. The proposed fix is simple and intuitive to the proposed issue, and demonstrates notable performance increase on the demonstrated experiments.

I have some curiosities around the experiments and the drawn conclusions. Most notably, the claim that "higher rank results in worse performance" would need more diverse experiments than training on a single regularization term. It is unclear to me why model performance can be disentangled from potential learning instability from the regularizer, or if this results from the loss "focusing too much" on the regularizer (instead of e.g. using a hinge loss, where features are thresholded to some minimum rank). If the authors have an explanation for why this is not a concern, or if there are alternative methods for rank promotion that can be tried which still degrade performance, this will greatly increase the strength of their claim given the lack of theoretical evidence. Since this is a more fundamental VQ fix rather than a specific CV application, it's more excusable that experiments are restricted to now older datasets like CIFAR10 and CelebA, but in the absence of theory, it would be notably strengthened with more modern examples (e.g. ImageNet).

---

> ### Author Rebuttal · Authors · 2025-07-31
>
> # Response to Reviewer enVD
> We greatly appreciate the reviewer’s thorough evaluation and insightful feedback. We address each of your points in detail below.
>
> ## Response to Q1
>
> We acknowledge the concern that KoLeo’s performance drop may not reflect true higher-rank incompatibility. To test this hypothesis, we conducted experiments with multiple alternative rank-promoting methods: Barlow Twins, VICReg, SingHinge (our hinge loss on singular values of covariance), and RankHinge (as suggested by the reviewer). All regularizers were applied on the unquantized latents. Due to the time limit, we only tested the CNN architecture with $f=4$. Detailed implementation is provided later in this response.
>
> **Table 1: CIFAR10 performance with different regularizers**
>
> | Regularizer Type|   Regularizer Weight |   Val Loss  |   Effective Dim  |
> |-----------------|--------------|-------------|------------------|
> | None|  -   |   0.081 |9 |
> | KoLeo   |   0.001  |   0.081 |9 |
> | KoLeo   |   0.01   |   0.082 |   10 |
> | KoLeo   |   0.1|   0.106 |   27 |
> | BarlowTwins |   0.0001 |   0.081 |   10 |
> | BarlowTwins |   0.001  |   0.093 |   31 |
> | BarlowTwins |   0.01   |   0.105 |   78 |
> | BarlowTwins |   0.1|   0.087 |   49 |
> | VIC |   0.1|   0.081 |9 |
> | VIC |   1  |   0.080 |   10 |
> | VIC |  10  |   0.082 |   11 |
> | VIC | 100  |   0.085 |   12 |
> | SingHinge(0.01) |   0.1|   0.080 |9 |
> | SingHinge(0.01) |   1  |   0.081 |9 |
> | SingHinge(0.01) |  10  |   0.089 |   49 |
> | SingHinge(0.01) | 100  |   0.129 |  111 |
> | SingHinge(0.1)  |   0.1|   0.080 |9 |
> | SingHinge(0.1)  |   1  |   0.081 |   10 |
> | SingHinge(0.1)  |  10  |   0.084 |   11 |
> | SingHinge(0.1)  | 100  |   0.085 |   12 |
> | RankHinge(64)   |   0.001  |   0.081 |9 |
> | RankHinge(64)   |   0.01   |   0.081 |9 |
> | RankHinge(64)   |   0.1|   0.080 |9 |
> | RankHinge(64)   |   1  |   0.081 |9 |
> | RankHinge(128)  |   0.001  |   0.080 |9 |
> | RankHinge(128)  |   0.01   |   0.082 |9 |
> | RankHinge(128)  |   0.1|   0.080 |9 |
> | RankHinge(128)  |   1  |   0.081 |9 |
>
> (Val Loss reports validation reconstruction loss only, excluding the regularization term. SingHinge($h$) applies a hinge loss on singular values with threshold $h$ (e.g., 0.01 or 0.1). RankHinge($r$) applies a hinge loss on feature rank with threshold $r$ (e.g., 64 or 128). Regularization weights were chosen to yield performance comparable to the no-regularization baseline.)
>
> From these results, we find a consistent trend: increasing the strength of KoLeo, Barlow Twins, VICReg, or SingHinge increases the effective dimension but also degrades performance once it grows beyond the baseline level (≈9). This pattern holds across very different gradient structures, suggesting the effect is not an artifact of KoLeo but a fundamental property of the VQ quantization bottleneck.
>
> RankHinge is a special case. While the codebook effective dimension remains flat (~9) regardless of weight, we want to emphasize that this is not due to insufficient hyperparameter tuning: the unquantized latents show clear growth in dimensionality, reaching levels similar to KoLeo (shown below). Even with regularization weights up to 1000, the effective dimension stays at 9 while validation loss worsens (not shown), underscoring a regularizer limitation rather than a tuning issue.
>
> **Table 2: RankHinge regularization effective dimension vs. unquantized effective dimension**
> | Regularizer Type   |   Regularizer Weight |   Val Loss  |   Effective Dim  |   Effective Dim (before Quant) |
> |----------------|--------------|-------------|------------------|--------------------------------|
> | None    | - |0.081 |  9 | 27 |
> | KoLeo   | 0.001 |0.081 |  9 | 19 |
> | KoLeo   | 0.01  |0.082 | 10 | 76 |
> | KoLeo   | 0.1   |0.106 | 27 |123 |
> | RankHinge(64)  | 0.001 |0.081 |  9 | 17 |
> | RankHinge(64)  | 0.01  |0.081 |  9 | 33 |
> | RankHinge(64)  | 0.1   |0.08  |  9 | 52 |
> | RankHinge(64)  | 1     |0.081 |  9 | 54 |
> | RankHinge(128) | 0.001 |0.08  |  9 | 20 |
> | RankHinge(128) | 0.01  |0.082 |  9 | 15 |
> | RankHinge(128) | 0.1   |0.08  |  9 |128 |
> | RankHinge(128) | 1     |0.081 |  9 |133 |
>
>
> ## Response to "it would be notably strengthened with more modern examples (e.g. ImageNet)"
> We appreciate the reviewer’s suggestion to include more modern datasets. We wanted to clarify that Section 5.2 includes experiments on ImageNet for our method and RQVAE. Here, we add comparison with vanilla VQVAEs and other regularizers on ImageNet:
>
> **Table 3: ImageNet-1k performance with different methods**
> | Type|   Regularizer Weight |   Val Loss  |   Effective Dim  |
> |-----------------|--------------|-------------|------------------|
> | BarlowTwins |   0.0001 |   0.173 |   14 |
> | BarlowTwins |   0.001  |   0.186 |   41 |
> | BarlowTwins |   0.01   |   0.205 |   91 |
> | BarlowTwins |   0.1|   0.182 |   58 |
> | VIC |   0.1|   0.172 |9 |
> | VIC |   1  |   0.164 |   11 |
> | VIC |  10  |   0.169 |   13 |
> | VIC | 100  |   0.18  |   20 |
> | SingHinge(0.01) |   0.1|   0.158 |9 |
> | SingHinge(0.01) |   1  |   0.168 |   11 |
> | SingHinge(0.01) |  10  |   0.181 |   63 |
> | SingHinge(0.01) | 100  |   0.247 |  137 |
> | SingHinge(0.1)  |   1  |   0.171 |   11 |
> | SingHinge(0.1)  |  10  |   0.183 |   15 |
> | SingHinge(0.1)  | 100  |   0.186 |   17 |
> | RankHinge(64)   |   0.001  |   0.173 |9 |
> | RankHinge(64)   |   0.01   |   0.173 |9 |
> | RankHinge(64)   |   0.1|   0.173 |9 |
> | RankHinge(64)   |   1  |   0.173 |9 |
> | RankHinge(128)  |   0.001  |   0.166 |9 |
> | RankHinge(128)  |   0.01   |   0.163 |9 |
> | RankHinge(128)  |   0.1|   0.168 |9 |
> | RankHinge(128)  |   1  |   0.166 |9 |
> | DCVQ(4x8d)  |   - |   0.115 |   20 |
> | DCVQ(8x8d)  |   - |   0.105 |   36 |
>
> Similar to CIFAR10, we observe that increasing the regularization strength leads to higher effective dimensions but also worse validation loss. RankHinge maintains a low effective dimension while showing no performance degradation.
>
> Please notice that the results presented here are of a different setting compared to Section 5.2. Here, we didn't train the models to converge due to time constraints. Instead, we stopped training after 21k steps, at which point the effective dimensions had stabilized. Another difference is the use of a smaller parameter size. We plan to include full convergence results in the camera-ready version.
>
> ## Response to Q2
>
> In the above tables, we explored a diverse set of regularization strategies, which can be grouped into three categories:
> - Embedding spreading (e.g., KoLeo): Encourages the embeddings to be spread out in the latent space.
> - Redundancy reduction methods (e.g., Barlow Twins, VICReg): These encourage feature decorrelation and variance, commonly used in self-supervised learning.
> - Covariance spectrum shaping (e.g., SingHinge): These directly regularize the singular values of the feature covariance matrix to promote higher rank.
> - Rank thresholding (e.g., RankHinge): Inspired by the reviewer’s suggestion, this penalizes low-rank representations only when effective dimension falls below a target threshold.
>
>
> ## Response to Q3
>
> We observe that using more subspaces generally helps preserve high-frequency details in the reconstructions (e.g., hair). However, increasing the dimensionality of each subspace beyond our recommended 8 dimensions often harms these details. Due to rebuttal constraints, we are unable to upload visual examples at this stage, but we will include detailed qualitative comparisons and reconstructions in the camera-ready version.
>
> ## Implementation Details for Regularizers
>
> For **Barlow Twins** and **VICReg**, which were originally designed for contrastive learning with augmented view pairs, data augmentation is not needed in our rank-promotion setting. Instead, we form input pairs by duplicating the same latent vector. All other hyperparameters follow the original codebase.
>
> ### Barlow Twins Loss
>
> Given normalized latent vectors $ x, y \in \mathbb{R}^{B \times D} $, where $B$ denotes the total number of patches across GPUs and $D$ the feature dimension, and letting $\text{BN}$ denote 1D batch normalization, the Barlow Twins loss is defined as:
>
> $$
> C = \frac{1}{B} \, \text{BN}(x)^\top \text{BN}(y) \\\\
> \mathcal{L}_{\text{Barlow}} = \sum_i (C_{ii} - 1)^2 + 0.0051 \sum_{i \neq j} C_{ij}^2
> $$
>
> ### VICReg Loss
>
> Given $x, y, B, D$ defined similarly, the VICReg loss combines variance and covariance regularization:
>
> The variance term:
>
> $$
> \mathcal{L}_{\text{var}}
> = \frac{1}{D} \sum_{i=1}^D \frac{\max(0, 1 - \sigma(x_{\cdot i})) + \max(0, 1 - \sigma(y_{\cdot i}))}{2}
> $$
>
> where $\sigma$ denotes the standard deviation.
>
> The covariance term:
>
> $$
> \mathcal{L}_{\text{cov}} = \frac{1}{D} \sum_{i \neq j} (\text{Cov}(x)_{ij}^2 + \text{Cov}(y)_{ij}^2)
> $$
>
> with $\text{Cov}(x) = \tfrac{1}{B-1}(x - \mu(x))^\top(x - \mu(x))$, where $\mu(x)$ is the batch mean.
>
> The total VICReg loss is then:
>
> $$
> \mathcal{L}_{\text{VICReg}} = \mathcal{L}_{\text{var}} + \tfrac{1}{25} \cdot \mathcal{L}_{\text{cov}}
> $$
>
> ### SingHinge
>
> For latent representations $z \in \mathbb{R}^{B \times D}$, let $s_i$ denote the singular values of $\text{Cov}(z)$. With a threshold $h$, the SingHinge loss is defined as
>
> $$
> \mathcal{L}_{\text{SingHinge}(h)} = \frac{1}{D} \sum_i \max(0, h - s_i)
> $$
>
> ### RankHinge
>
> The intuition behind RankHinge is to ensure that at least $r$ singular values are necessary to capture 99% of the variance. To achieve this, we penalize the case where the first $r-1$ singular values already explain $\geq 99\%$ of the variance. Formally, let
>
> $$
> E_{r-1} = \sum_{i=1}^{r-1} \frac{s_i}{\sum_j s_j}
> $$
>
> denote the cumulative variance explained by the top $r-1$ singular values. The RankHinge loss is then defined as
>
> $$
> \mathcal{L}_{\text{RankHinge}(r)} = \max(0,\, E_{r-1} - 0.99)
> $$
>
> **References:**
>
> [1] Barlow Twins: Self-Supervised Learning via Redundancy Reduction
>
> [2] VICReg: Variance-Invariance-Covariance Regularization for Self-Supervised Learning

---

> ### Author Response · Authors · 2025-08-05
>
> Thank you again for taking the time to review our work. Since the discussion phase is nearing its end, we wanted to check whether the additional extension and ablation experiments we included have helped to address your concerns. Please let us know if any further clarification would be helpful.

---

> ### Author Response · Authors · 2025-08-07
> **Follow-up on Rebuttal Response**
>
> Dear Reviewer enVD,
>
> We understand you may have a busy schedule, and we truly appreciate the time you’ve taken to review our submission. As the rebuttal period is closing in the next 24 hours, we just wanted to check in—if you have any further comments or questions, we’d be eager to respond if needed.
>
> Thank you again for your time and consideration.
>
> Best regards,
>
> The authors

---

> > ### Comment · Reviewer_enVD · 2025-08-07
> >
> > Apologies for the delay, thank you for the detailed rebuttal. This was exactly the kind of analysis I was looking for and greatly appreciate the time taken for clarifications and more comprehensive experiments on short notice.
> >
> > I have raised my score accordingly. It is an interesting paper that gives a new new light on the feature dimension/collapse discussion, that still has many interesting and ongoing questions.

---

> > > ### Author Response · Authors · 2025-08-08
> > >
> > > Thank you for the thoughtful follow-up and for reconsidering the score. I’m glad the additional clarifications and experiments were helpful, and I truly appreciate your engagement with the work.

---

### Decision · Program_Chairs · 2025-09-17

**Decision:**

Accept (poster)

**Comment:**

This paper identifies and presents a study of dimensional collapse in VQ-GANs. Applying simple rank regularization leads to poor performance, so this paper proposes an alternative method for spanning the space better. The reviewers all praised the paper for the importance of the tackled problem, clarity, and simplicity. Most concerns raised in the reviews have been addressed in the rebuttal. For all the above reasons, I recommend accepting this paper as poster contribution to NeurIPS 2025.